EMBO
Molecular Medicine

# The evolving genetic landscape of telomere biology disorder dyskeratosis congenita

Hemanth Tummala [1,2✉], Amanda J Walne[1], Mohsin Badat[1,2], Manthan Patel [1], Abigail M Walne[1], Jenna Alnajar[1], Chi Ching Chow [1], Ibtehal Albursan [1], Jennifer M Frost [1], David Ballard [3], Sally Killick[4], Peter Szitányi[5], Anne M Kelly[6], Manoj Raghavan [7], Corrina Powell[8], Reinier Raymakers[9], Tony Todd [10], Elpis Mantadakis[11], Sophia Polychronopoulou[12], Nikolas Pontikos[13], Tianyi Liao[1], Pradeep Madapura [1], Upal Hossain[1,2], Tom Vulliamy [1] & Inderjeet Dokal [1,2]

## Abstract

**Dyskeratosis congenita (DC) is a rare inherited bone marrow failure syndrome, caused by genetic mutations that principally affect telomere biology. Approximately 35% of cases remain uncharacterised at the genetic level. To explore the genetic landscape, we conducted genetic studies on a large collection of clinically diagnosed cases of DC as well as cases exhibiting features resembling DC, referred to as 'DC-like' (DCL). This led us to identify several novel pathogenic variants within known genetic loci and in the novel X-linked gene, *POLA1*. In addition, we have also identified several novel variants in *POT1* and *ZCCHC8* in multiple cases from different families expanding the allelic series of DC and DCL phenotypes. Functional characterisation of novel *POLA1* and *POT1* variants, revealed pathogenic effects on protein-protein interactions with primase, CTC1-STN1-TEN1 (CST) and shelterin subunit complexes, that are critical for telomere maintenance. *ZCCHC8* variants demonstrated ZCCHC8 deficiency and signs of pervasive transcription, triggering inflammation in patients' blood. In conclusion, our studies expand the current genetic architecture and broaden our understanding of disease mechanisms underlying DC and DCL disorders.**

**Keywords** Dyskeratosis Congenita; Telomeres; *POLA1*; ncRNAs
**Subject Categories** DNA Replication, Recombination & Repair; Genetics, Gene Therapy & Genetic Disease; Musculoskeletal System

See also: G Guenechea & NW Meza

## Introduction

Dyskeratosis congenita (DC; MIM 305000) is an inherited syndrome of bone marrow (BM) failure, classically characterized by a triad of muco-cutaneous features i.e., abnormal skin pigmentation, nail dystrophy, and oral leucoplakia. In addition, DC is associated with pleiotropic abnormalities affecting the dental, gastrointestinal, genitourinary, neurological, ophthalmic, pulmonary, skeletal, and vascular systems (Tummala et al, 2022a, 2022b; Dokal et al, 2022). BM failure represents a significant cause of mortality in individuals with DC, and they are also predisposed to cancer, pulmonary and liver complications. While some DC individuals respond to androgen therapy, the only curative option for haematopoietic defects is haematopoietic stem cell transplantation using fludarabine-based conditioning protocols, but this cannot improve the non-haematopoietic tissues affected by this condition (Tummala et al, 2022a, 2022b).

DC can manifest as X-linked recessive, autosomal dominant, or autosomal recessive subtypes. Currently, nineteen DC genes/loci have been identified: *DKC1*, *TERC*, *TERT*, *NOP10*, *NHP2*, *TINF2*, *TCAB1*, *USB1*, *CTC1*, *RTEL1*, *ACD*, *PARN*, *NAF1*, *ZCCHC8*, *NPM1*, *MDM4*, *RPA1*, *DCLRE1B* and *TYMS-ENOSF1* (Tummala et al, 2022a, 2022b; Revy et al, 2023; Niewisch et al, 2023). The germline variants found thus far principally impair the function of the telomerase holoenzyme (TERT and *TERC*) in terms of maturation, stability, trafficking, and the safeguarding of telomeres against external DNA damage and unfavourable or adverse recombination events during cell division. While telomere defects are central, non-telomere defects such as ribosomal abnormalities also play a role in specific DC subtypes (Tummala et al, 2022a, 2022b). In addition, disease severity may increase in succeeding generations due to the inheritance of short telomeres, known as the anticipation phenomenon (Vulliamy et al, 2004).

[1]Centre for Genomics and Child Health, Blizard Institute, Faculty of Medicine and Dentistry, Queen Mary University of London, Newark Street, London E12AT, UK. [2]Barts Health NHS Trust, London, UK. [3]Department of Analytical, Environmental & Forensic Sciences, Kings College London, Franklin-Wilkins Building, Stamford Street, London SE1 9NH, UK. [4]Department of Haematology, Royal Bournemouth Hospital NHS Foundation Trust, Bournemouth BH7 7DW, UK. [5]Department of Paediatrics and Inherited Metabolic Disorders, First Faculty of Medicine, Charles University and General University Hospital in Prague, Ke Karlovu 2, 128 08 Praha 2, Prague, Czech Republic. [6]Cambridge University Hospitals, Cambridge Biomedical Campus, Cambridge CB2 0QQ, UK. [7]Clinical Haematology, Queen Elizabeth Hospital, Edgbaston, Birmingham B15 2TH, UK. [8]Clinical Genetics, Birmingham Women's and Children's NHS Foundation Trust, Birmingham B15 2TG, UK. [9]University Medical Center Utrecht, 3508 GA Utrecht, The Netherlands. [10]Department of Haematology, Royal Devon and Exeter Hospital, Exeter EX2 5DW, UK. [11]Department of Pediatrics' University General Hospital of Alexandroupolis, Democritus University of Thrace Faculty of Medicine, 6th Kilometer Alexandroupolis-Makris, 68 100 Alexandroupolis, Thrace, Greece. [12]Department of Pediatric Hematology-Oncology, Aghia Sophia Children's Hospital, Athens, Greece. [13]Institute of Ophthalmology, Faculty of Brain Sciences, University College London, Gower St, London WC1E 6BT, UK.
✉E-mail: h.tummala@qmul.ac.uk

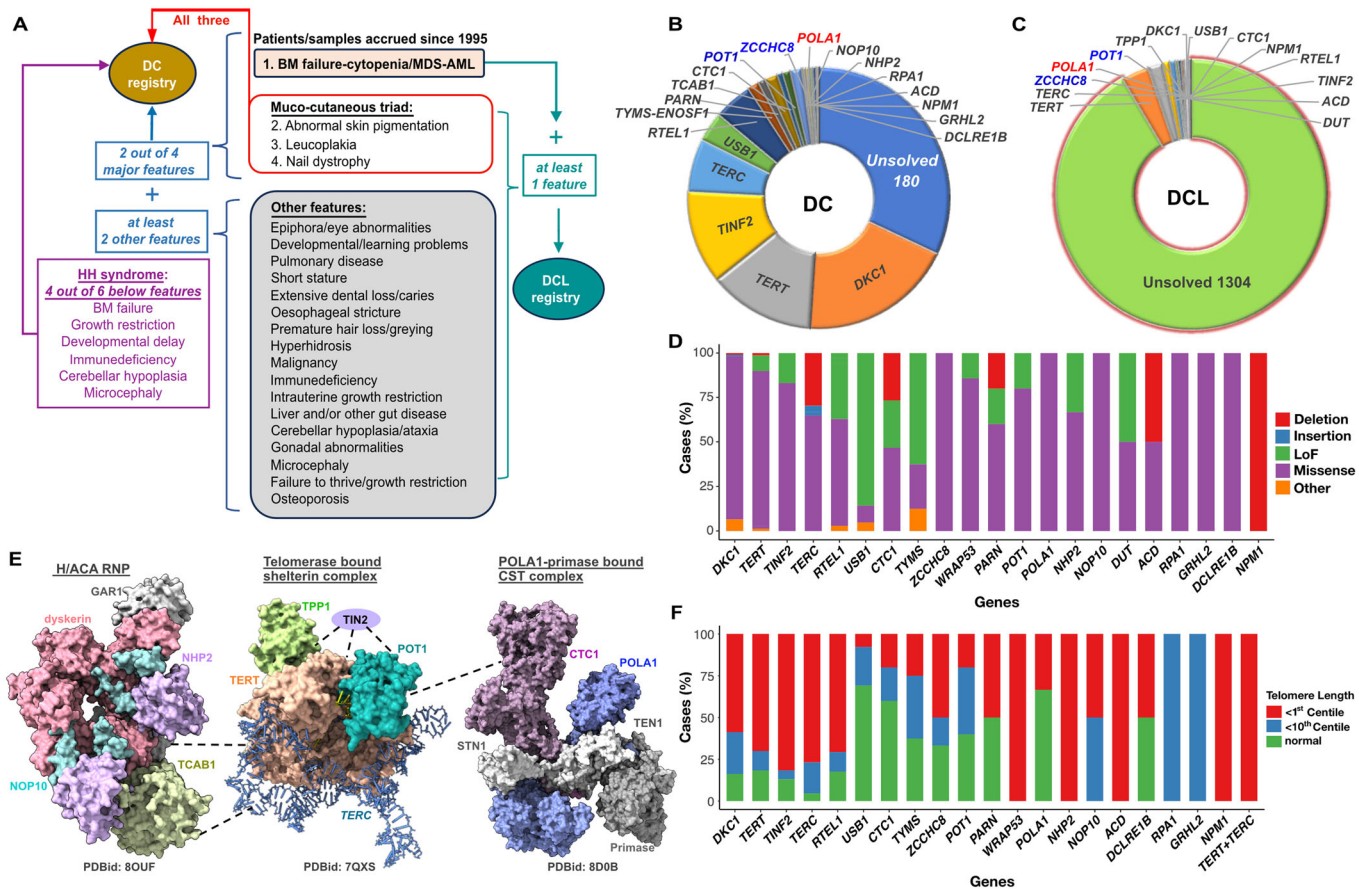

**Figure 1. Genetic landscape of dyskeratosis congenita (DC; DC-Like, DCL).**

(A) Patient samples were initially categorized based on haematological abnormality, bone marrow (BM) failure-cytopenia/MDS-AML (myelodysplasia-acute myeloid leukaemia). Patients in DC registry include: (i) those with all muco-cutaneous features/triad. (ii) Those with at least two of the four (BM failure, abnormal skin pigmentation, leucoplakia, nail dystrophy,) major features together with at least two "other features". (iii) Those with at least four out of the six features of Hoyeraal Hreidarsson (HH) syndrome. Patients in DCL registry include those with BM failure and at least one of the "other features" or one muco-cutaneous feature. HH refers to 'Hoyeraal Hreidarsson'. (B, C) The genetic architecture reveals characterized cases with gene variants that overlap in clinically classified cases of DC and DCL phenotypes. (Dataset EV1, Table 1). (D) Types of variants observed across all affected individuals in known DC genes and in the new loci POLA1. The variant profile includes 336 missense, 58 LoF, 26 deletions, and 3 insertions. In total, 13 variants fell into other category, where predicted protein coding was either duplication or uncertain, were identified across the affected individuals. (E) Structure of the telomerase and telomere regulating complexes highlighting the known genes and 1 new gene POLA1 bound to CST complex that are mutated in the DC and DCL cohort. Dotted lines indicated interactions of protein subunits across these complexes. (F) Telomere lengths measured by either mm-QPCR, southern blotting or HT-STELA in DC and DCL cases.

We established an international registry that specifically focused on collecting samples and clinical information about individuals diagnosed with DC and dyskeratosis congenita-like (DCL) disorders since 1995. The cases included in this cohort (some of which have been previously published and contributed to our fundamental understanding of DC; Dataset EV1) were gathered through a systematic initiative, and other ad hoc referrals, originating from both institutions within the United Kingdom and international sources. This approach has allowed for a diverse and extensive collection of cases, contributing to the comprehensive nature of our study. The minimal clinical criteria for diagnosis of DC include the presence of at least two out of the four major features (nail dystrophy, abnormal skin pigmentation, leucoplakia and BM failure) and at least two of the other extra-haematopoietic features known to occur in DC and its

severe variant Hoyeraal Hreidarsson syndrome (Fig. 1A; Vulliamy et al, 2006; Dokal, 2001). Whereas the DCL patients present with BM failure and with at least one extra-haematopoietic feature seen in DC but not sufficient to be classified as DC (Fig. 1A; Dokal, 2011).

In this report, we describe comprehensive analysis of the genetic landscape of one of the largest collections of DC and DCL patients worldwide. We provide detailed functional studies relating to three specific disease genes in which either no DC/DCL patients had been described previously (POLA1) or because only few families have been previously associated with that gene (POT1 and ZCCHC8). Detailed studies on these novel variants in these three genes significantly expand our understanding of the molecular mechanisms that occur beyond canonical long non-coding RNA TERC in causing disease features in DC.

**Table 1. Novel germline *POLA1*, *POT1* and *ZCCHC8* variants identified in DC and DCL cases.**

| Code | Case | Gene | DNA | Zygosity | Protein | gnomAD | CADD |
|---|---|---|---|---|---|---|---|
| DC 029 | 1 | *POLA1* | c.1846A>G | Hemi | p.Arg616Gly | NR | 25.3 |
| DC 029 | 2 | *POLA1* | c.1846A>G | Hemi | p.Arg616Gly | NR | 25.3 |
| DCL 060 | 2 | *POLA1* | c.3437 A > G | Hemi | p.Tyr1146Cys | NR | 27.6 |
| DCL 060 | 4 | *POLA1* | c.3437 A > G | Hemi | p.Tyr1146Cys | NR | 27.6 |
| DCL 793 | 5 | *POLA1* | c.1609 C > A | Hemi | p.Leu537Ile | NR | 18.43 |
| DCL 389 | 6 | *POLA1* | c.1846A>G | Hemi | p.Arg616Gly | NR | 25.3 |
| DC 236 | 7 | *POT1* | c.1294 C > T | Het | p.Arg432Ter | NR | 38 |
| DCL 440 | 8 | *POT1* | c.1255 T > C | Het | p.Tyr419His | 1.32E-05 | 24.2 |
| DC 448 | 9 | *POT1* | c.728 G > A | Het | p.Ser243Asn | NR | 20.2 |
| DC 441 | 10 | *POT1* | c.437 C > T | Het | p.Pro146Leu | NR | 20.3 |
| DCL441 | 11 | *POT1* | c.437 C > T | Het | p.Pro146Leu | NR | 20.3 |
| DC 441 | 12 | *POT1* | c.437 C > T | Het | p.Pro146Leu | NR | 20.3 |
| DC 460 | 13 | *POT1* | c.1651 G > A | Het | p.Gly551Arg | NR | 33 |
| DC 460 | 14 | *POT1* | c.1507 T > G/c.1651 G > A | Comp het | p.Cys503Gly/p.Gly551Arg | NR/NR | 26.7; 33 |
| DC 460 | 15 | *POT1* | c.1507 T > G/c.1651 G > A | Comp het | p.Cys503Gly/p.Gly551Arg | NR/NR | 26.7; 33 |
| DCL 699 | 16 | *ZCCHC8* | c.586 G > A | Het | p.Glu196Lys | NR | 32 |
| DC 411 | 17 | *ZCCHC8* | c.337 G > A | Het | p.Glu113Lys | NR | 26.1 |
| DC 116 | 18 | *ZCCHC8* | c.508 G > A | Het | p.Gly170Arg | NR | 28.5 |
| DC 271 | 19 | *ZCCHC8* | c.658 G > C/c.508 G > A | Comp Het | p.Val220Leu/p.Gly170Arg | NR/NR | 22.9; 28.5 |
| DCL 879 | 20 | *ZCCHC8* | c.196 G > A | Het | p.Glu66Lys | NR | 29 |
| DCL 215 | 21 | *ZCCHC8* | c.508 G > C | Het | p.Gly170Arg | NR | 27.9 |
| DCL 215 | 22 | *ZCCHC8* | c.508 G > C | Het | p.Gly170Arg | NR | 27.9 |
| DCL 61 | 23 | *ZCCHC8* | c.551 G > A | Het | p.Gly184Glu | NR | 27.3 |
| DCL 61 | 24 | *ZCCHC8* | c.551 G > A | Het | p.Gly184Glu | NR | 27.3 |

*NR* not reported.
Gene accession numbers for *POLA1*: NM_016937; *POT1*: NM_015450 and *ZCCHC8*: NM_017612.

# Results

## Spectrum of genetic variants in patients with DC and DCL features

We have undertaken analysis of the germline landscape of DC (461 families; $n = 190$ exomes; and $n = 157$ targeted gene panel, $n = 51$ whole genome sequencing on affected cases) and DCL families (1566 families; $n = 228$ exomes; $n = 764$ targeted gene panel (Appendix Table S1), $n = 13$ whole genome sequencing on affected cases). The genetic landscape comprises of a significant subset of genes overlapping between DC and DCL phenotypes (Fig. 1B,C; Dataset EV1) and the majority of the putative causal variants are typical missense or LoF alleles, with higher frequency of missense variants observed in the first reported gene for this condition, *DKC1* (Fig. 1D). While we intentionally omitted cases lacking a definitive diagnosis of syndromic features found in other inherited BMF syndromes like Fanconi anaemia (FA; MIM: 60713 characterised by abnormal chromosomal breakage), Shwachman–Diamond Syndrome (SDS; MIM: 607444, associated with exocrine pancreatic insufficiency), and Diamond Blackfan anaemia (DBA; MIM: 105650, characterised by reticulocytopenia or bone marrow erythroblastopenia), we identified a subset of patients harbouring variants in genes known to be mutated in these recognised syndromes and these have been excluded ($n = 62$ families) from both registries. Our analysis has unveiled novel pathogenic variants in previously known DC genes (Dataset EV1) that encode for the telomerase complex (TERT and *TERC*), telomerase ribonucleoprotein (RNP) complex (Dyskerin, NOP10, and TCAB1), shelterin component (TPP1, TIN2 and POT1) and telomere replication component (RTEL1, DCLREIB and CTC1). More than ~2/3 of genes identified in DC and DCL patients from our cohort are known to function in telomere biology (Fig. 1E) and telomere lengths for majority of these patients are either very short (< 1st centile) or short (< 10th centile) with exception to *USB1* and *CTC1* patients (Fig. 1F). Furthermore, the genetic landscape of DC and DCL includes *DUT*, that is involved in nucleotide metabolism similar to *TYMS-ENOSF1* (Tummala et al, 2022a, 2022b) and *GRHL2* which encodes a transcriptional regulator of *TERT* (Chen et al, 2010; Walne et al, 2016). In this study, we report an evolving genetic landscape of DC, comprising an allelic series with a spectrum of novel variants identified in *POT1* and *ZCCHC8*, and in a previously undescribed locus *POLA1* (Fig. 1; Dataset EV1). All these novel variants are evolutionarily conserved (Fig. EV1A–C) and predicted to be damaging as determined by combined annotation-dependent depletion (CADD) scores (Rentzsch et al, 2019; Table 1). Below, we present studies demonstrating the functional consequences of these newly identified variants in

*POLA1*, *POT1* and *ZCCHC8* and expand the current genetic mechanisms underlying DC and DCL disorders.

## *POLA1* variants impact POLA1 catalytic efficiency

Focusing on families with only affected males to enrich for X-linked recessive disease variants, we analysed whole exome sequencing data from 120 uncharacterised families, excluding those with an autosomal dominant pattern of inheritance (Appendix Table S2). We enriched for potentially pathogenic alleles by selecting variants that were not reported on the gnomAD (version 3.1.2) aggregation databases. In this way, we found 11 genes that had novel variants in two or more unrelated individuals (Appendix Tables S3 and S4), but only one gene (*POLA1*) in which novel X-linked variants (p.Arg616Gly, p.Tyr1146Cys) were segregating in two unrelated multiplex families (DC 029, DCL 060; Fig. 2A). The affected males from these two families had hypoplastic bone marrow and pancytopenia (Table 2). The maternal uncle in family DC 029 died of lung fibrosis and myelodysplasia (MDS) and their paternal great grandmother had leukaemia. Further screening of additional samples from males with bone marrow failure for variants in *POLA1* revealed two additional cases who had variants in *POLA1*. The first shared the same variant that we had identified in family DC 029 (p.Arg616Gly; DCL 389). This simplex case (DCL 389) presented with cytopenia and features of hypoplastic MDS in the marrow (Table 2). The second carried the variant p.Leu537Ile (DCL 793) and had a hypoplastic bone marrow and pancytopenia. The identified *POLA1* variants segregate with disease in the affected individuals and obligate carriers.

We initially assessed the functional impact of these *POLA1* variants by studying the X-chromosome inactivation pattern (XCIP) in the blood cells of female *POLA1* carriers. The mothers that are carriers of these novel missense *POLA1* variants showed complete skewing of X inactivation pattern in families DC 029 and DCL 060 (Fig. 2B). This is highly suggestive of a damaging effect, giving rise to a selective advantage to bone marrow cells expressing the wild-type allele. It indicates a critical role for POLA1 in bone marrow cell proliferation/survival and is reminiscent of previous observations in X-linked dyskeratosis congenita due to variants in *DKC1* (Vulliamy et al, 1997; Heiss et al, 1998).

POLA1 belongs to a family of DNA polymerases: α, δ and ε (Baranovskiy et al, 2016a, 2016b). The majority of DNA replication is conducted by DNA polymerases δ and ε, but DNA polymerase α (POLA1), in complex with primase, initiates the replication process by synthesizing Okazaki-fragments on both chromosomal DNA strands (Burgers and Kunkel, 2017; Michael et al, 2000; Baranovskiy et al, 2018). The POLA1 variants that we have discovered are located within the catalytic domain and may impact the efficiency of pairing the template DNA with the primer (Fig. 2C; Baranovskiy et al, 2018; Baranovskiy et al, 2016a, 2016b; Baranovskiy et al, 2016a, 2016b). To investigate this, we introduced GFP-tagged POLA1 (wild type and variants) into HEK293 cells and conducted co-immunoprecipitation (Co-IP) using GFP-TRAP beads (Fig. 2D). Subsequently, we utilized the protein complexes bound to the GFP-TRAP beads to perform an in vitro primer extension assay. In this assay, we employed a fluorescently labelled RNA primer that was annealed to the corresponding DNA template (Fig. 2E), and we introduced CD437 toxin, a specific POLA1 inhibitor known to selectively inhibit POLA1 activity in increasing concentrations as

previously described (Han et al, 2016). The results of this experiment revealed that the identified POLA1 variants exhibited diminished activity in extending the DNA–RNA primer, when compared to the wild type. Additionally, we evaluated the impact of a common POLA1 variant, p.P496S, on POLA1 activity in extending the DNA–RNA primer and found no significant defect (Fig. 2F,G). It should be noted that none of the identified missense variants impacted the cellular localization of POLA1; when expressed in HeLa cells, they predominantly appeared nuclear (Fig. EV1D,E).

## *POT1* variants impact POT1 binding to single-strand telomeric DNA

POT1 is a component of the shelterin complex, which consists of six subunits (TRF1, TRF2, TPP1, POT1, TIN2, RAP1), and plays a negative regulatory role in telomere elongation by acting as a cap at the ends of the telomeres (Palm and de Lange, 2008). Previous studies have implicated germline variants in *POT1* in specific telomere syndromes, such as Coats plus and idiopathic pulmonary fibrosis (Takai et al, 2016; Kelich et al, 2022). In this study, we present novel *POT1* variants identified in several DC and DCL individuals from five families (DC 236, DCL 440, DC 448, DC 441, DC 460; Fig. 3A). In family DC 448, the variant was found to be de novo, while in families DC 441 (P10, P11, P12) and DC 460 (P14, P15) the variants segregated in autosomal dominant and recessive patterns, respectively (Fig. 3A). All cases exhibited haematological abnormalities ranging from T-cell deficiency leading to immune defects and hypocellular bone marrows (Table 3). Patients 11 and 12 in family DC 441 presented with nail dystrophy and abnormal skin pigmentation (Fig. 3B–D). The identified *POT1* variants are novel and appear to affect the OB-fold DNA-binding domains, the Holliday junction resolvase-like (HJRL) motif and the TPP1 binding motif in OB3-fold domain of POT1 (Fig. 3E). To assess the impact of these POT1 variants on binding to telomeric single-stranded DNA (ssDNA), we performed an electrophoretic mobility shift assay (EMSA), following previously established protocol (Schratz et al, 2021). Using transcription-coupled translation system in vitro, the generated POT1 wild-type and variant proteins were verified by western blotting (Fig. 3F) and subjected to EMSA with Cy3 labelled telomeric ssDNA probe. This revealed that all pathogenic POT1 variants exhibited reduced binding to telomeric ssDNA in comparison to the wild type and specifically the POT1 C-terminal variants identified in DC 236 and DC 460 families that reside in TPP1 binding motifs (Sekne et al, 2022) have completely failed to form specific higher-order POT1-telomeric ssDNA binding complexes (Fig. 3G,H).

## *POLA1* and *POT1* variants disrupt CST and shelterin complex interactions

POLA1-primase facilitates the completion of the complementary C-strand through its interaction with the CST (CTC1-STN1-TEN1) complex, which possesses ssDNA binding capabilities and regulates the length of the 3' G-overhang. In addition, the CST complex interacts with POT1 and TPP1 to prevent telomerase from binding to telomeres (Zaug et al, 2022; Zaug et al, 2021; Chen et al, 2012). Several studies have reported that biallelic pathogenic variants in *CTC1* and *STN1* are implicated in the development of Coats plus

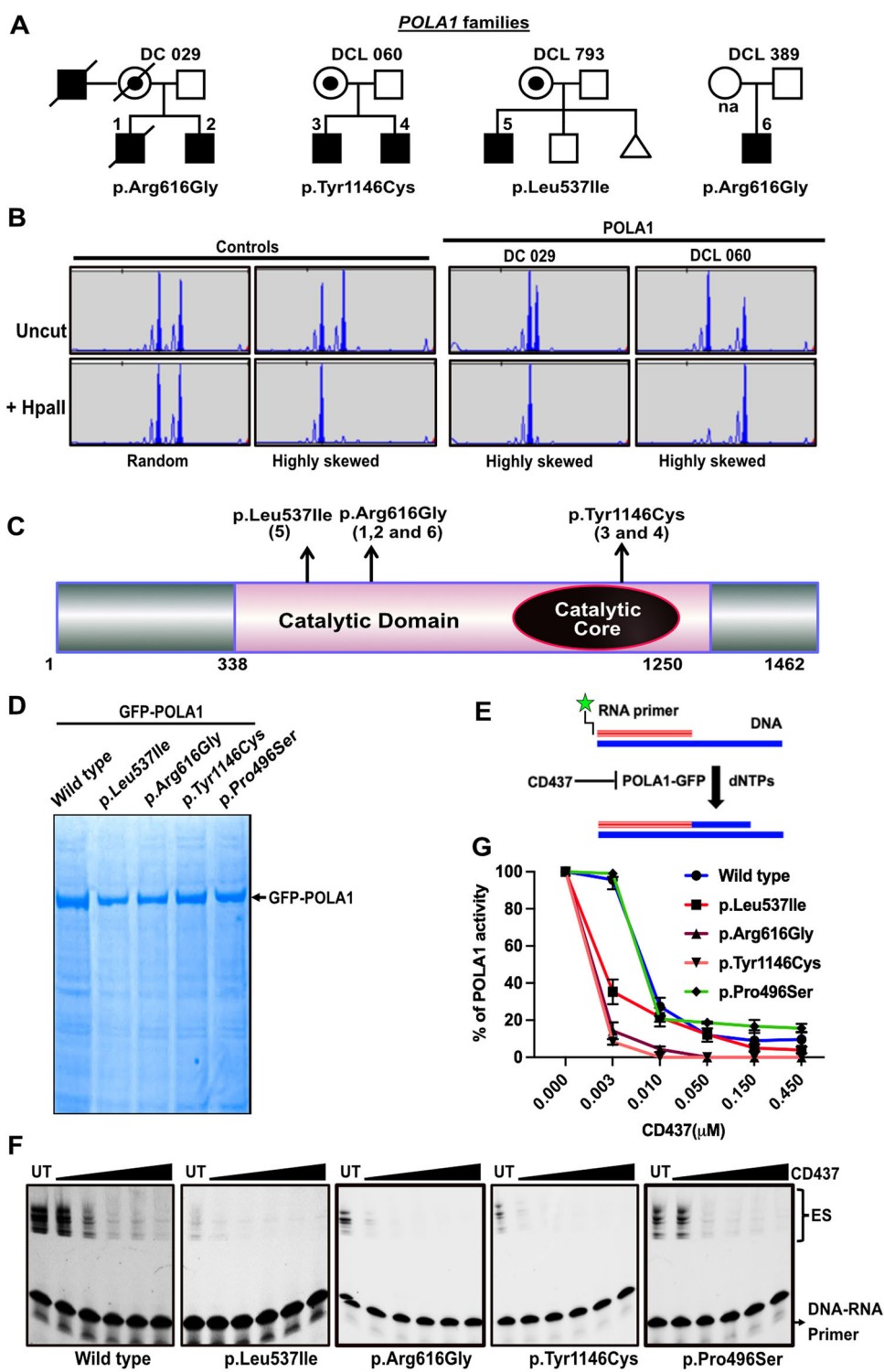

syndrome and, in rare instances, dyskeratosis congenita (Anderson et al, 2012; Walne et al, 2013; Simon et al, 2016; Passi et al, 2020; Polvi et al, 2012). Notably, these patients carrying *CTC1* or *STN1* pathogenic variants did not consistently exhibit reduced leucocyte telomere length (Kelich et al, 2022; Zaug et al, 2022). In line with this, telomere length patterns were found to be variable in patients

with pathogenic *POLA1* and *POT1* variants when measured using the high-resolution telomere length measurement technique STELA (single telomere length analysis; Fig. 4A–C; Norris et al, 2021). STELA revealed an increase in telomere truncations in the genomic DNA of whole blood (< 2 kb) for some patients (Fig. 4A). Others displayed longer telomere length products (> 7 kb) when

Figure 2. *POLA1* variants impact POLA1 catalytic efficiency.

(A) Family pedigrees for individual cases displaying novel variants in a hemizygous state in *POLA1*. Circle with dot—carrier. (B) Analysis of maternal X-Chromosome inactivation pattern (XCIP). *HpaII* treatment digests the active allele, allowing amplification exclusively from the inactive allele. (C) Schematic diagram illustrating the presence of variants in the catalytic domain of POLA1. (D) Coomassie staining reveals proteomes of GFP-POLA1 wild-type and variant proteins enriched through GFP-TRAP purification from 293T protein lysates expressing the respective proteins. (E) RNA-DNA duplexes utilized in the in vitro primer extension assay. (F) In vitro POLA1 primer extension activity conducted in the presence of the specific POLA1 inhibitor CD437 toxin. The primer extension activity is visibly inhibited with increasing concentrations of CD437 toxin (0.003 μM, 0.01 μM, 0.05 μM, 0.15 μM and 0.45 μM). (G) POLA1 activity determination based on intensity values of extended species (ES) in each lane, relative to the untreated lane, which is set as 100% POLA activity in each group. Error bars represent SE of mean intensities from $n = 3$ independent replicate experiments. Source data are available online for this figure.

**Table 2. Clinical features of individuals with hemizygous *POLA1* variants.**

| DC/DCL patient | DCL 060 3 | DCL 060 4 | DC 029 1 | DC 029 2 | DCL 389 6 | DCL 793 5 |
|---|---|---|---|---|---|---|
| **Features** | | | | | | |
| Gender | M | M | M | M | M | M |
| Country/ethnic origin | USA | USA | UK | UK | UK | Argentina |
| Age at investigation (yrs) | 21 | 18 | 36 | 30 | 40 | 11 |
| Parents consanguineous | N | N | N | N | N | N |
| Skin pigment abnormalities | N | N | N | N | N | N |
| Nail dystrophy | N | N | N | Y | N | N |
| Leukoplakia | N | N | N | N | N | N |
| Grey hair | N | N | Y | Y | N | N |
| Haematological abnormalities | Y[a] | Y[b] | Y[c] | Y[d] | Y[e] | Y[f] |
| Immune defects | N | N | N | N | N | N |
| Other features | N | N | Y[g] | Y[h] | Y[i] | Y[j] |

*M* male, *F* female, *Y* yes, *N* normal/no, *?* unknown.
[a]Pancytopenia with macrocytosis; [b]hypocellular bone marrow with pancytopenia; [c]hypocellular bone marrow with pancytopenia; [d]hypocellular bone marrow with pancytopenia; [e]hypocellular myelodysplasia; [f]reduced bone marrow cellularity, reduced megakaryocytes and thrombocytopenia.
[g]skeletal (hip), hepatic and renal abnormalities, died of multi-system abnormalities aged 45 yrs; [h]skeletal (hip) abnormalities, renal impairment; [i]emphysema and pulmonary fibrosis progressing to fatal respiratory failure at age 41 yrs; [j]low birthweight, growth restriction, deafness, hypogonadism.

compared to control blood at reduced intensity (Fig. 4A–C). Furthermore, we observed a subtle reduction in POT1 protein levels along with a lower molecular weight product specifically in the lymphoblastoid cells established from *POT1* patients when compared to control (Fig. 4D). Sequencing the cDNA from these patient cells did not reveal any mis-spliced transcripts. This suggests that the observed lower molecular weight POT1 protein could be a direct result of an unknown post-translational modification occurring as a consequence of *POT1* variants. Further an increase in ATR-CHK1-P53 signalling axis is observed in *POT1* patient cells treated with ATM kinase inhibitor KU55933 and DNA-PK inhibitor NU7026, implying enhanced ATR-mediated activation of DNA damage response, that is a characteristic feature of *POT1* deficient cells (Dench and de Lange 2007; Fig. 4D). In *POT1* patient cells we specifically observed an increase in the telomeric ssDNA-binding protein RPA1 aka RPA70 (Fig. 4D). In cell proximity ligation assay (PLA) unveiled notably strong

interactions between 53BP1, and TRF2 as well as between TRF2 and RPA70 in *POT1* patient cells (Fig. 4E,F). This increase of PLA foci between 53BP1 and TRF2 signifies dysfunctional telomeres, while RPA70 loading denotes increase in ssDNA G-overhangs at telomeres. Studies elsewhere have also demonstrated that POLA1 inhibition leads ssDNA accumulation and RPA1 exhaustion causing DNA replication catastrophe (Ercilla et al, 2020). Collectively, our results suggest that *POT1* and *POLA1* variants may exert variable telomere end fill-in reactions causing increased loading of RPA70 and 53BP1 at dysfunctional telomeres.

Structural cryo-EM studies uncovered that the CST complex forms a physical association with POLA1-primase, enhancing the efficiency of primer synthesis (Cai et al, 2022; He et al, 2022). Given the suboptimal priming of DNA–RNA primers by POLA1 due to pathogenic variants (Fig. 2G), we conducted experiments to examine the interaction between the POLA1-primase complex and the CST complex. To accomplish this, we expressed constructs of the CST complex tagged with epitopes (MYC-CTC1, FLAG-STN1 and TEN1-HA), in conjunction with GFP-tagged wild-type and variant forms of POLA1, in 293T cells. Subsequent Co-IP by GFP-TRAP beads revealed weak interaction between POLA1 variants and PRIM1, PRIM2A, as well as the components of the CST complex, in comparison to the wild-type control and the commonly found p.P496S POLA1 variant (Fig. 4G). Recently, the CST-POT1/TPP1 complex, established through interactions between POT1 and the CTC1 subunit of CST, was described (Cai et al, 2024). We investigated these interactions, by expressing epitope-tagged Myc-POT1 wild type and variants in conjunction with FLAG-TPP1 and His-CTC1 in 293T cells. Analysis of the Co-IP complexes using MYC-TRAP beads revealed loss of interaction between TPP1 and the POT1p.Arg432* variant when compared to wild-type control. Furthermore, a majority of the analysed *POT1* variants exhibited an enhanced affinity for binding to CTC1 when compared to wild type (Fig. 4H). Collectively, these studies provide valuable insights into the crucial physical association of the CST complex with POLA1-primase, and the POT1-TPP1 complexes, underscoring the impact of pathogenic variants in both *POLA1* and *POT1* on these interactions, which can have significant implications on telomeric DNA replication and stability.

### *ZCCHC8* variants exert ZCCHC8 deficiency with no significant impact on *TERC* levels

Previously, a monoallelic missense variant in *ZCCHC8* (p.Pro186-Leu) was documented to co-segregate with low levels of mature *TERC* in a single family, where individuals exhibited short telomeres, pulmonary fibrosis, or bone marrow failure (Gable

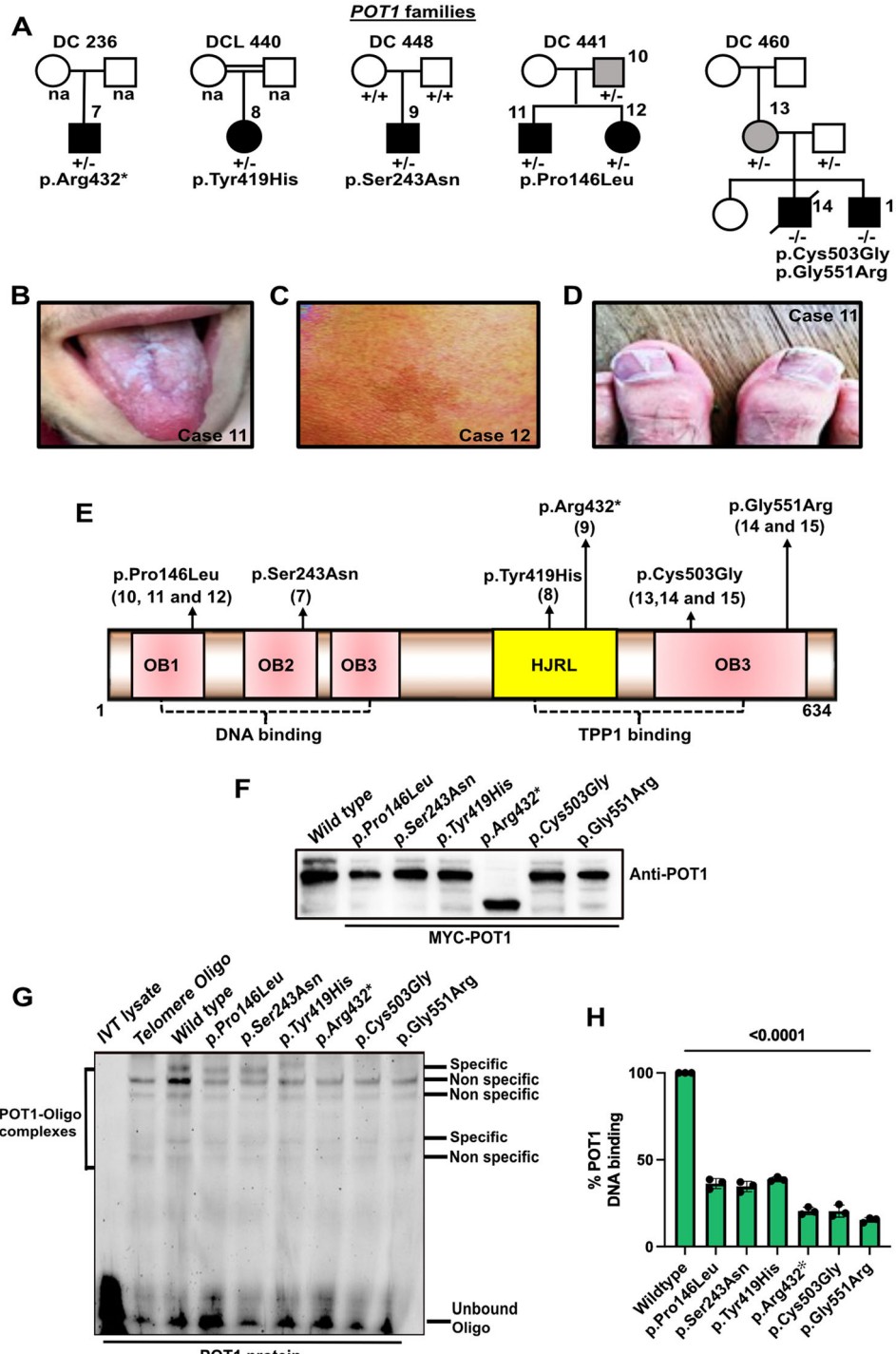

**Figure 3. POT1 variants disrupt POT1 binding to single-strand telomere DNA.**

(A) Family pedigrees for individual cases displaying novel variants in heterozygous (+/−) state in *POT1*. Light grey square/circle—parent with mild phenotype. (B–D) Clinical photographs of *POT1* patients displaying leucoplakia on the tongue and nail dystrophy in patient 11, as well as abnormal skin pigmentation in patient 12 from family DC 441. (E) A schematic diagram illustrating the location of variants within the conserved domains of POT1. (F) Immunoblot of in vitro translated products demonstrating stable expression of missense mutant POT1 at comparable levels to wild type, with specificity confirmed by the POT1 antibody. (G) Electrophoretic mobility shift assay (EMSA) for wild type and mutant POT1, assessing its binding capacity to a telomere oligonucleotide. (H) Mean intensity of telomere oligonucleotide binding for POT1 variants relative to wild type was determined by measuring the intensity optical density of both 'specific' indicated bands on the gel. The data presented with error bars representing standard deviation from $n = 3$ independent EMSA experiments conducted with in vitro translation reactions. '*P*' values are calculated by using ordinary one-way ANOVA test. Source data are available online for this figure.

**Table 3. Clinical features of individuals with *POT1* variants.**

| DC/DCL patient | DC 236 7 | DCL 440 8 | DCL 448 9 | DC 441 10 | DC 441 11 | DC 441 12 | DC 460 13 | DC 460 14* | DC 460 15 |
|---|---|---|---|---|---|---|---|---|---|
| **Features** | | | | | | | | | |
| Gender | M | F | M | M | M | F | F | M | M |
| Country/ethnic origin | France | UK-Pakistan | Czech Republic | UK | UK | UK | UK | UK | UK |
| Age at investigation (yrs) | 6 | 15 | 10 | 52 | 16 | 12 | 59 | 28 | 20 |
| Parents consanguineous | N | Y | N | N | N | N | N | N | N |
| Skin pigment abnormalities | N | N | N | Y (40 s) | N | N | N | N | N |
| Nail dystrophy | Y | N | Y (4 yrs) | N | N | Y (12 yrs) | N | N | N |
| Leukoplakia | N | N | N | N | Y | N | N | N | N |
| Grey hair | N | N | N | N | N | Y (17 yrs) | Y | Y (13 yrs) | Y (16 yrs) |
| Haematological abnormalities | N[a] | Y[b] | Y[c] | Y[d] | Y[e] | Y[f] | ?[g] | ?[h] | Y[i] |
| Immune defects | Y | N | N | N | N | N | ? | ? | ? |
| Other features | Y[j] | Y[k] | Y[l] | Y[m] | Y[n] | Y[o] | Y[p] | Y[q] | Y[r] |

*M* male, *F* female, *Y* yes, *N* normal/no, *?* unknown.
*Patient deceased; [a]low T-cells and poor mitogen response; [b]hypocellular bone marrow with thrombocytopenia; [c]hypocellular bone marrow with leucopenia and thrombocytopenia; [d]normal blood count but macrocytosis and hypocellular bone marrow; [e]hypocellular bone marrow with leucopenia and thrombocytopenia; [f]hypocellular bone marrow with leucopenia and thrombocytopenia; [g]full details not available; [h]full details not available, patient deceased; [i]hypocellular bone marrow with leucopenia and thrombocytopenia; [j]recurrent respiratory infections, dysphagia, cerebellar abnormality; [k]learning problems, mal-aligned teeth, epilepsy, recurrent chest infections; [l]intra-uterine growth restriction, hepato-splenomagyly, phimosis, lung fibrosis with vessel abnormalities, retinal haemorrhages; [m]asthma, chronic pelvic/bladder pain; [n]thin teeth, attention deficit hyperactivity disorder (ADHD); [o]ADHD; [p]at time of writing being investigated for shortness of breath/respiratory disease; [q]born at 33 weeks, learning problems, joint hyper-mobility, liver cirrhosis and pulmonary fibrosis leading to death at age 28 yrs; [r]low birthweight, learning problems, dry skin, recurrent respiratory symptoms.

et al, 2019). Here, we report novel germline *ZCCHC8* variants in multiple individuals across seven independent families (Fig. 5A) displaying features associated with DC or DCL (as detailed in Table 4). Notably, we have observed a recurrent variant (p.Gly170Arg) in four independent patients from three families (DC 116, DC 271, and DCL 215). Specifically, in DCL 215 family both siblings presented with this recurrent ZCCHC8 variant (p.Gly170Arg), while both parents when tested appear to be wild type. The chances of both non-twin siblings developing a de novo disease-causing variant is unlikely. Instead, we suspect that one of the parents, who carries the variant, is a germline or somatic mosaic, as such events have been previously observed in DC (Vulliamy et al, 1999). In patient 19 from the DC 271 family, biallelic variants in *ZCCHC8* (p.Gly170Arg and p.Val220Leu) were detected. All patients had severe bone marrow failure, with patients 23 and 24 exhibiting myelodysplasia characterized by refractory cytopenia and multilineage dysplasia (Table 4). Whole blood telomere lengths as determined by STELA revealed lengths of 50th percentile for patient 20, 10th percentile for patient 17, and below the 1st percentile for patients 22, 16 and 19 (Fig. 5B).

ZCCHC8 serves as a scaffolding protein that interacts with the RNA helicase MTR4 and the RNA binding protein RBM7 to construct the NEXT complex (Fig. 5C). In our patient cohort, akin to a previously identified *ZCCHC8* variant (p.Pro186Leu), all the pathogenic variants we identified are also situated within the dimerization domain of ZCCHC8 (Fig. 5C,D). This domain plays a pivotal role in the assembly of the higher-order NEXT complex (Puno and Lima (2022); Gerlach et al, 2022). Through this intricate mechanism, the multimeric NEXT complex facilitates the degradation of various types of nuclear RNAs, including both poly(A)+ and poly(A)-, and short and long non-sequence-specific RNAs, by recruiting the exosome components EXOSC10 and DIS3 (Wu et al, 2020; Schmid and Jensen, 2018).

Moreover, ZCCHC8 is recognized for its involvement in the transcriptional repression of transposable elements within embryonic stem cells through its interaction with MPP8, a constituent of the chromatin modifier HUSH complex (Garland et al, 2022). To investigate the impact of the identified *ZCCHC8* variants we ectopically expressed GFP and MYC-tagged ZCCHC8 in ZCCHC8-3F-mAID HeLa-TIR1 cells (Gockert et al, 2022), as treatment with indole acetic acid (IAA) acutely depleted endogenous ZCCHC8 by 3 h time point (Fig. EV2A), therefore only allowing us to detect expression of proteins from ectopically expressed plasmid constructs. This revealed that none of the identified missense variants have impacted the cellular localization of ZCCHC8 and they predominantly appeared nuclear (Fig. EV2B), although the expression of variant ZCCHC8 protein appears to be compromised when compared to wild type (Fig. 5E).

Previously, both human and murine cell lines that are devoid of ZCCHC8 function showed decrease in *TERC* transcripts, encompassing both the precursor and mature forms (Gable et al, 2019). However, it remains unclear whether these processing defects in *TERC*, observed in these cells, are a consequence of impaired functionality within the nuclear exosome targeting (NEXT) complex, which promotes the degradation of non-coding transcripts without poly (A) tails. Therefore, we analysed *TERC* levels in patient RNA samples carrying pathogenic *ZCCHC8* variants. Although an increase in levels of *ZCCHC8* expression is detected (Fig. 5F), no significant change in the level of immature *TERC* transcripts (characterized by 3' extended oligo A tails) is observed in patients' whole blood RNA when compared to control samples (Fig. 5G). Notably, we are able to effectively distinguish immature *TERC*, in patients RNA samples with biallelic *PARN* variants, which is consistent with the previous reports using this assay (Fig. 5G; Moon et al, 2015; Tummala et al, 2022a, 2022b; Tummala et al, 2015). Furthermore, the levels of immature *TERC* appear to be only

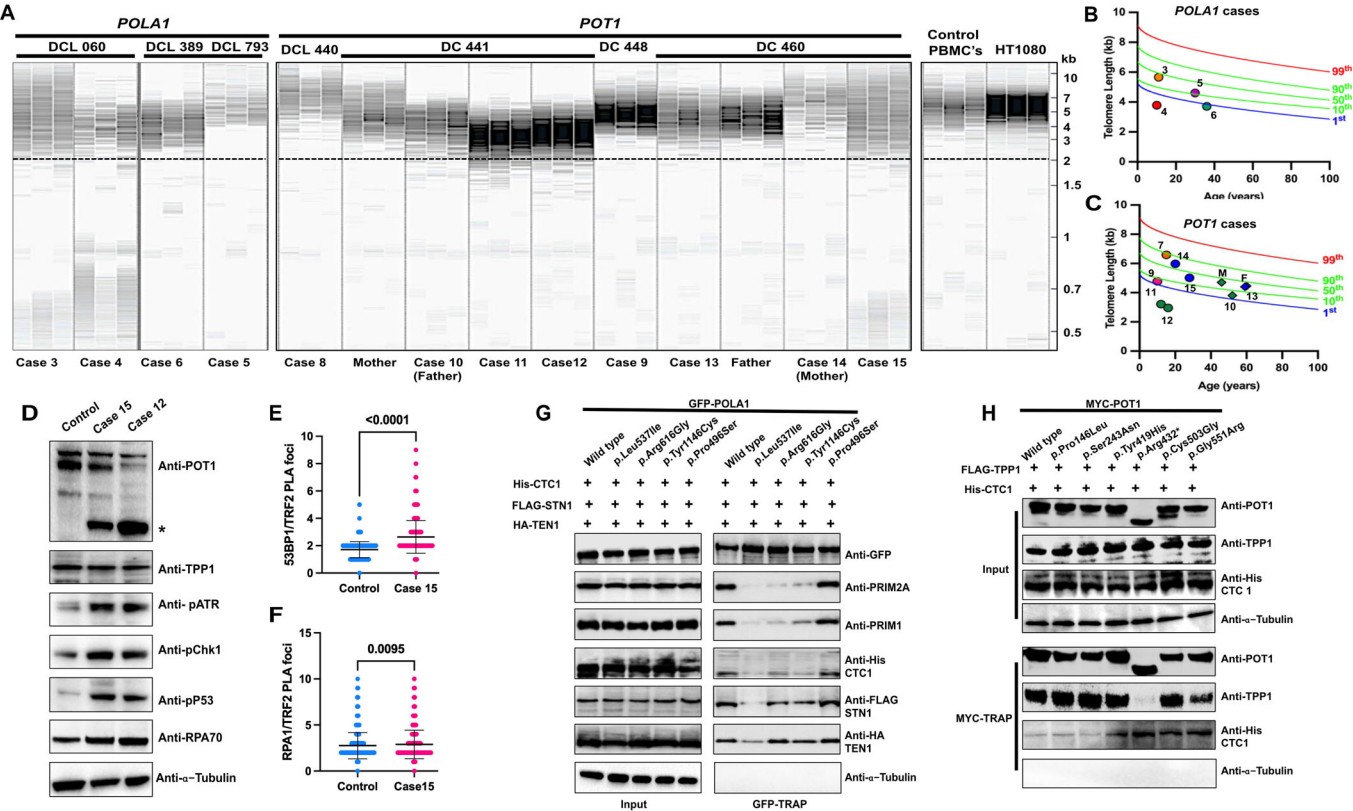

**Figure 4. POLA1 and POT1 variants disrupt telomere integrity and induce genome instability.**

(A) HT-STELA data, used to characterize telomere length variation in DNA extracted from peripheral blood samples of the patient cohort, revealed truncated telomeres (marked by a dotted line at 2 kb) as well as less amplified higher-sized telomeric products in some POT1 and POLA1 patients. (B, C) HT-STELA data from POLA1 and POT1 patients, plotted alongside predicted percentiles of unaffected individuals ($n = 4730$) based on age. (D) Western blot analysis of protein lysates demonstrating increased levels of DNA damage response proteins in POT1 patient cells compared to the control, with α-tubulin serving as a loading control. *Indicates lower molecular weight POT1 protein (possibly cleaved or degraded) product specifically observed in patient cells. (E, F) In-cell confocal analysis of proximity ligation assay (PLA) dots in control and POT1 patient cells, confirming interactions between the 53BP1 and RPA70 proteins with the telomere protein TRF2. Data represent means ± standard deviation, $n = 2$ independent experiments capturing 800 cells in each condition, with 'P' values determined by two-tailed unpaired 't' test as reported in the graphs. (G, H) Western blot analysis of proteins in the input and pull-down of cell lysates expressing corresponding proteins using agarose GFP-TRAP and MYC-TRAP beads. Source data are available online for this figure.

mildly affected in 293T cells treated with *ZCCHC8* siRNA (Fig. 5H,I). To directly assess the impact of *ZCCHC8* pathogenic variants on 3' extension of *TERC* transcripts, we performed whole blood RNA sequencing (RNA-seq) on patient cells. Consistent with the qPCR data (Fig. 5G), the RNA-seq analysis has also revealed a very low enrichment of 3' extended *TERC* species, while significant increase in the reads of *ZCCHC8* transcripts is observed in these patients (Fig. 5J,K). This apparent discrepancy in *TERC* regulation between our study and those previously reported by Gable et al, (2019), could be due to the complete absence of ZCCHC8 in their genetically knockout ZCCHC8−/− HCT116 cell lines. In contrast, patient fibroblasts carrying the *ZCCHC8* variant, as reported by Gable et al, 2019, show mild levels of ZCCHC8 protein expression, that is similar to our RNAi knockdown studies (Fig. 5H,I). However, acute depletion of ZCCHC8 with IAA treatment in ZCCHC8-3F-mAID HeLa cells, significantly increased both 3' polyadenylated and total forms of *TERC*, as well as Telomeric Repeat containing RNA-*TERRA* (Fig. EV2C–E). Altogether, these results indicate that inherited ZCCHC8 deficiency has minimal impact on *TERC* regulation in the patient cells.

## Inherited ZCCHC8 deficiency triggers pervasive transcription and inflammation

ZCCHC8 plays a crucial role in the targeted degradation of abundant transcripts originating from open chromatin regions, such as promoters, promoter flanking regions, and transcription factor binding sites (Wu et al, 2020; Schmid and Jensen, 2018; Garland et al, 2022). Cells with depletion of ZCCHC8 exhibited increased levels of promoter upstream transcripts (PROMPTs), enhancer RNAs (eRNAs), and spliced snoRNA host gene (SNHG) long non-coding RNAs (lncRNAs; Garland et al, 2022). To have a comprehensive view of the global transcriptomic changes, we analysed ribo-depleted whole blood transcriptome datasets from *ZCCHC8* patients (Fig. 6A,B; Guo et al, 2022). This revealed dysregulation of various RNA species, that include several small and long ncRNAs, PROMPTs and eRNAs in the patients' blood samples in comparison to the control group ((log2-fold change RPKM, false discovery rate [FDR] <0.05); Fig. 6C–E; Table EV1). Notably, the C/D box containing snoRNAs arising from snoRNA host gene loci with predominant enrichment in growth arrest

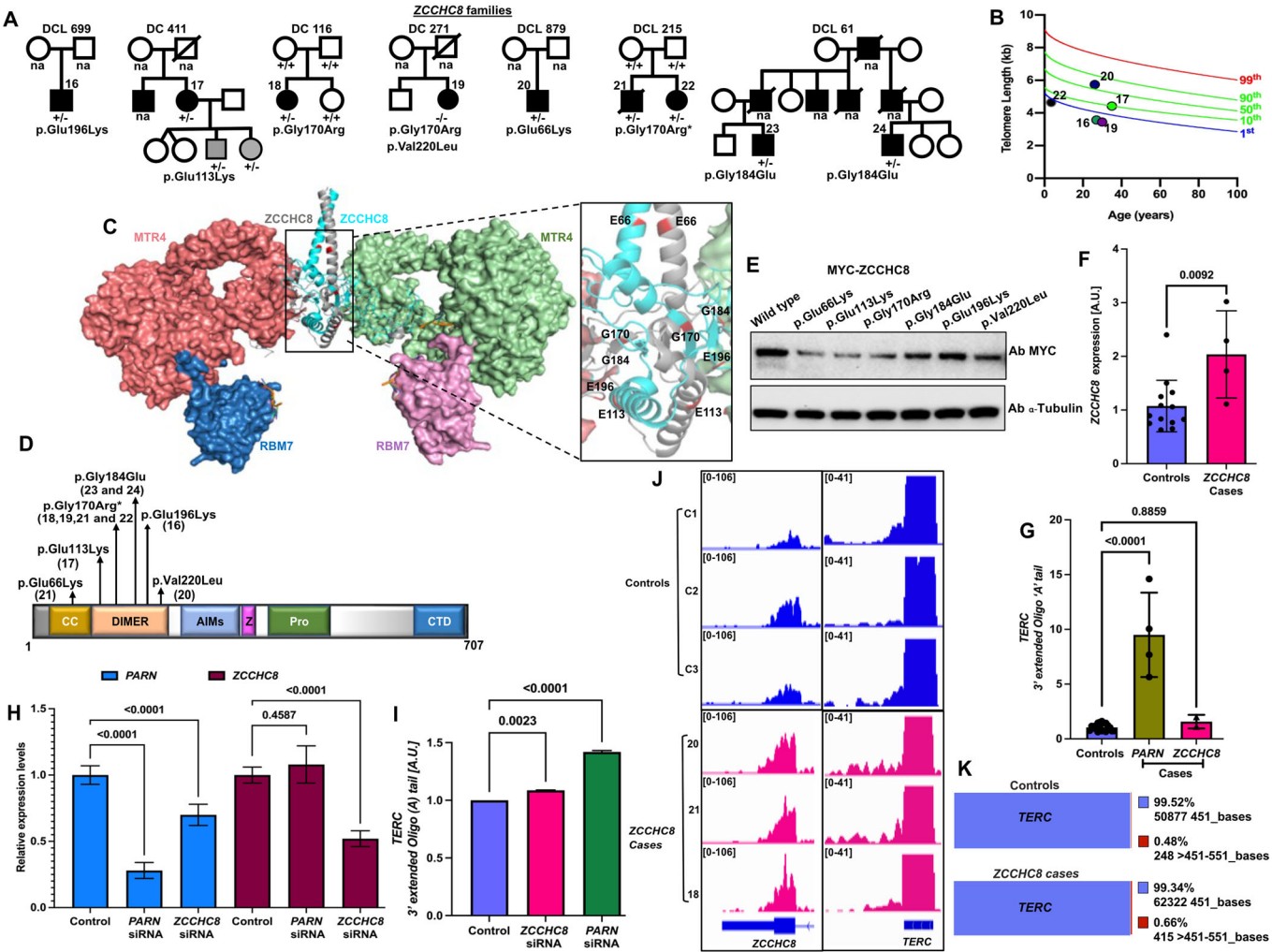

**Figure 5. Germline ZCCHC8 deficiency does not affect TERC.**

(A) Family pedigrees for individual cases displaying novel variants in a heterozygous (+/−) state in *ZCCHC8*. Light grey square/circle –children with mild phenotype in family DCR 411. (B) HT-STELA data from *ZCCHC8* patients, plotted alongside predicted percentiles of unaffected individuals based on age. (C) ZCCHC8 variants mapped onto the Cryo-EM structure (PDB id: 7Z4Y) of the NEXT complex, which includes ZCCHC8, MTR4, and RBM7 subunits. (D) Schematic diagram illustrating the location of variants within the conserved dimerization domain of ZCCHC8. (E) Immunoblotting of HeLa cell lysates expressing MYC-tagged wild-type and variant forms of ZCCHC8. (F) Whole blood RNA samples from *ZCCHC8* cases demonstrate increased levels of ZCCHC8 expression compared to unrelated controls. All gene expression levels are normalized to *GAPDH* or *TFRC*. Data represent means ± standard deviation, $n = 3$, with P values determined by one-way ANOVA were reported on the graph. (G) Oligo-dT$_{(20)}$-primed mature *TERC* RNA transcripts are distinguished from random hexamer-primed cDNA obtained from RNA samples from *ZCCHC8* and *PARN* patients' blood (the box represents the mean, and the whiskers represent the standard deviation). Data represent means ± standard deviation, $n = 2$, with P values determined by one-way ANOVA as reported on the graph. (H) Expression levels of *ZCCHC8* and *PARN* transcripts after transfection with the corresponding siRNA in HeLa cells. Expression is relative to the non-target control in each group. Data represent means ± standard deviation, $n = 2$, with P values determined by one-way ANOVA as reported on the graph. (I) Oligo-dT$_{(20)}$-primed mature *TERC* RNA transcripts are distinguished from random hexamer-primed cDNA obtained from RNA samples treated with siRNA targeting *ZCCHC8* and *PARN* genes. Data represent means ± standard deviation, $n = 2$, with P values determined by one-way ANOVA as reported on the graph. (J) Genome browser read coverage plots from IGV viewer of RNA-seq reads encompassing *ZCCHC8* and *TERC* loci in *ZCCHC8* cases and controls. (K) Percentage of mature (451 bp) and immature (> 451 bp) *TERC* transcripts in both controls and *ZCCHC8* cases obtained from Deseq2 filtered reads. Source data are available online for this figure.

stimulator 5 (*GAS5*) was observed in *ZCCHC8* patient's blood (Fig. 6D,E). Considering the collaboration between the NEXT and HUSH complexes in controlling the expression of retrotransposons such as long interspersed nuclear element-1 (*LINE1* or *L1TEs*) and endogenous retroviruses (ERV), we also focused our analysis on these TEs in these patient blood transcriptomes (Fig. EV3A,B). We noted significant upregulation (log10fold change RPKM, FDR < 0.05) of specific *ERVs* and *L1TEs* in *ZCCHC8* patients' blood (Fig. EV3A,B). The observed increase in *L1TEs* (totalling 8533) in

the blood of *ZCCHC8* patients was found to involve full-length *L1TE* loci across the genome (Fig. 6F), and this phenomenon was observed on a chromosome-wide scale with no distinction between evolutionarily young (*L1PA6-L1PA1*) or old *L1TEs* when compared to the control group (Fig. EV3C). To further validate these findings, we conducted RT-PCR on patient RNA samples using primers that target the *GAS5* and *L1TE* (*ORF1* and *ORF2*) regions. Consistent with the RNA-seq data, *ZCCHC8* patients exhibited significant upregulation of both *GAS5* and *L1TE* encoding *ORF1* and *ORF2*

**Table 4.  Clinical features of individuals with *ZCCHC8* variants.**

| DC/DCL patient | DCL 699 16 | DCR 411 17 | DC 166 18 | DC 271 19 | DCL 879 20 | DCL 215 21 | DCL 215 22 | DCL 161 23* | DCL 161 24* |
|---|---|---|---|---|---|---|---|---|---|
| Variant | c.586 G > A | c.337 G > A | c.508 G > A | c.508 G > A/ c.658 G > C | c.196 G > A | c.508 G > C | c.508 G > C | c.551 G > A | c.551 G > A |
| Protein | p.G196K | p.G113K | p.G170R | p.G170R/ p.V220L | p.E66K | p.G170R | p.G170R | p.G184E | p.G184E |
| CADD | 32 | 26.1 | 28.5 | 28.5/22.9 | 29 | 27.9 | 27.9 | 27.8 | 27.8 |
| **Features** | | | | | | | | | |
| Gender | M | F | F | F | M | F | M | M | M |
| Country/ethnic origin | Netherlands | UK | Turkey | Hong Kong | UK | Greece | Greece | Canada | Canada |
| Age at investigation (yrs) | 27 | 35 | 16 | 30 | 26 | 3.5 | 1.5 | ? | ? |
| Parents consanguineous | N | N | N | N | ?N | N | N | N | N |
| Skin pigment abnormalities | Y | Y (7 yrs) | Y (9 yrs) | Y (11 yrs) | ?N | N | N | ? | ? |
| Nail dystrophy | N | Y | Y (1 yr) | Y (18 yrs) | ?N | N | N | ? | ? |
| Leukoplakia | N | N | Y (13 yr) | Y (18 yrs) | | N | Y | ? | ? |
| Hair loss/greying | Y | Y | N | N | ?N | N | N | ? | ? |
| Haematological abnormalities | Y[a] | Y[b] | Y[c] | Y[d] | Y[e] | Y[f] | Y[g] | Y MDS (RCMD) | Y MDS (RCMD) |
| Immune defects | N | N | N | N | ?N | N | N | ? | ? |
| Other features | Y[h] | Y[i] | N | Y[j] | ?N | N | Y[k] | ? | ? |

*F* female, *M* male, *Y* yes, *N* normal/no, *?* unknown.
[a]Hypocellular bone marrow with dysplasia, leucopenia and thrombocytopenia; [b]hypocellular bone marrow with thrombocytopenia; [c]thrombocytopenia;
[d]hypocellular bone marrow with anaemia and leucopenia; [e]hypocellular bone marrow with trilineage pancytopenia; [f]megablastoid bone marrow with reduced magakaryocytes, anaemia and thrombocytopenia; [g]hypocellular bone marrow with pancytopenia, died 10 months post-BMT of graft versus host disease.
[h]pulmonary fibrosis; [i]recurrent gingivitis, dental and hip problems; [j]dental problems; [k]severe skin toxicity post-BMT. *Cases are cousins as their fathers are half-brothers related by common father.

transcripts when compared to the control group and other genotypes of DC patients (Fig. 6G–I). Based on these transcriptional outcomes, gene ontology pathway analysis on differentially expressed genes (Fig. 6J) indicated a significant upregulation of pathways related to ribosome biogenesis, ncRNA processing, and DNA metabolism (Fig. 6K) in the blood of *ZCCHC8* patients. It should be noted that the upregulation of ribosome biogenesis is reminiscent of previous findings that were reported in whole blood transcriptome analysis of inherited bone marrow failure syndromes including DC (Walne et al, 2021). Conversely, there was a notable downregulation of pathways associated with myeloid activation and immune effector process in these patients (Fig. 6L). The major biological pathway determined by metascape analysis revealed for upregulation of genes involved in pro-inflammatory cascades such as type II interferon, cytokine and NF-κB signalling in *ZCCHC8* patients' blood in comparison to the controls (Figs. 6M and EV3D,E). In summary, our findings suggest that inherited ZCCHC8 deficiency enhances the pervasive transcription of genome that involve several lncRNAs (*GAS5*, and *L1TEs*) and drives pro-inflammatory signalling cascades, which may contribute to underlying syndromic disease features of DC and DCL subtypes.

# Discussion

Establishment of DC and DCL patient registry comprising clinical and biological data has facilitated research that has led to

improvements in our understanding of these condition(s), and patient care. By collecting and analysing data from many patients with DC and DCL phenotypes, over the years we have gained insights into the natural history of the disease, identified disease genetic variants, evaluated treatment approaches, and developed a better understanding of the overall impact of the condition. In this study, by analysing the largest series of DC and DCL families assembled to date, we shed light on the intricate genetic patterns underlying these disorders and showcase molecular similarities by identifying novel pathogenic variants within known loci, such as *POT1* and *ZCCHC8* in both DC and DCL subtypes. Furthermore, our genetic analysis also revealed pathogenic variants in a new locus *POLA1*, highlighting the evolving genetic landscape in DC and DCL subtypes (Fig. 1). Previously, germline variants in *POLA1* have been documented in cases of syndromic X-linked reticulate pigmentary disorder (XLRPD; MIM: 312040; Starokadomskyy et al, 2016) and X-linked intellectual disability (XLID; MIM: 301030; Van Esch et al, 2019). XLRPD is primarily characterized as an autoimmune deficiency syndrome, where patients commonly display NK cell maturation defects along with skin hyperpigmentation (Starokadomskyy et al, 2019). It should be noted that missense and splice site variants reported in XLID are located outside the catalytic domain of POLA1, unlike the catalytic variants identified in this study (Fig. 2C). Moreover, none of the previously reported XLID cases have been associated with bone marrow failure. While bone marrow failure combined with reticulate pigmentation is generally considered a defining

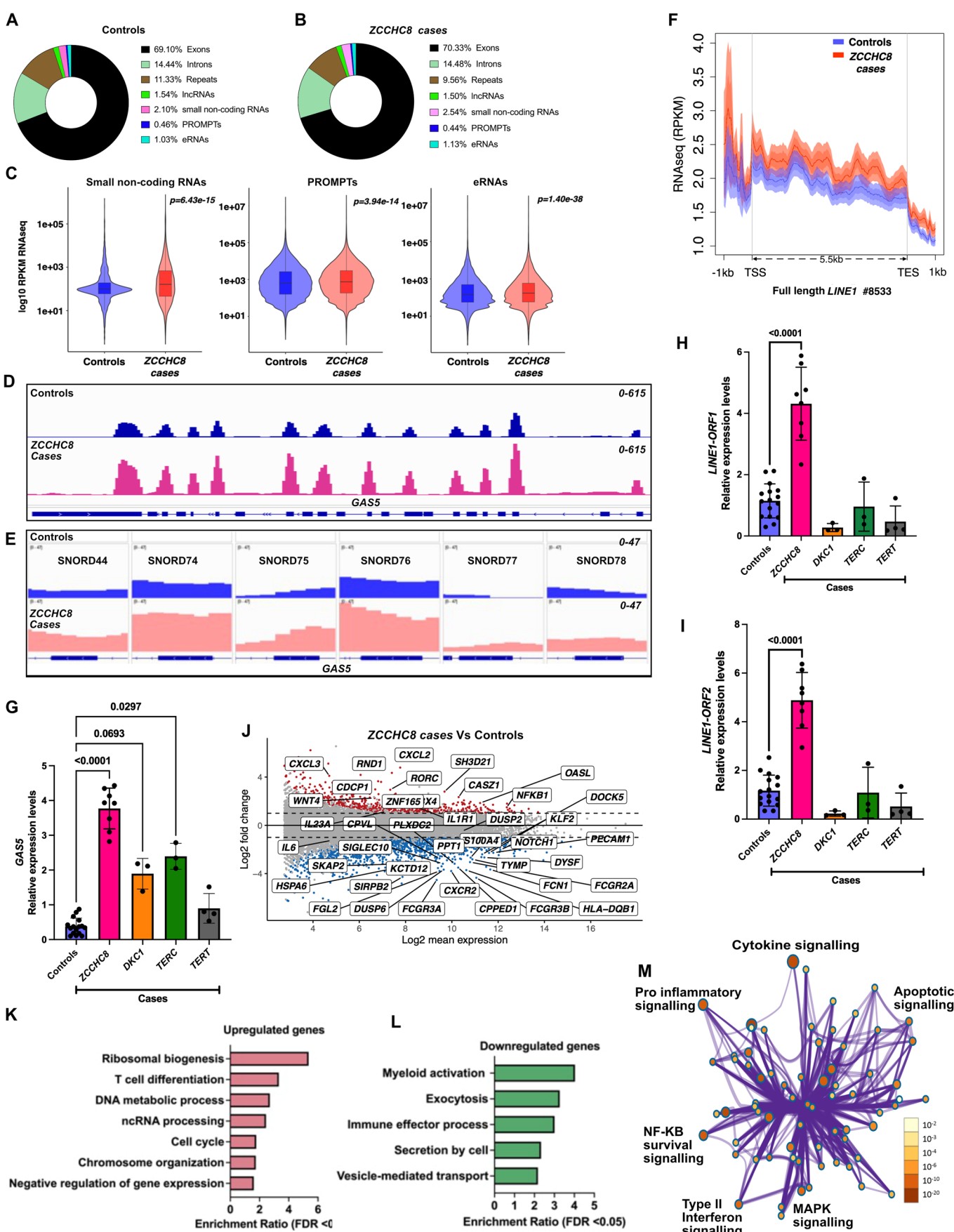

◀

**Figure 6. Global transcriptome analysis reveals non-coding RNA dysregulation and inflammation in *ZCCHC8* patients' blood.**

(A, B) Transcriptome analysis illustrating the RNA read coverage of various RNA species throughout the genome in both Control and *ZCCHC8* patients' whole blood RNA samples. (C) Alterations in the regulation of small non-coding RNAs, PROMPTs, and enhancer RNAs in control subjects compared to *ZCCHC8* patients. *p* values for all the violin/box plots were calculated using the pairwise two-sided multi-comparison Dunn test, a post hoc test. The box represents interquartile range of RPKM values of the expression of individual elements compared between control (*n* = 3) and *ZCCHC8* cases (*n* = 3). Violin-box plots indicate the median, bounds indicate the 25th and 75th percentiles, and whiskers limit show 1.5× interquartile range. (D) Genome browser read coverage plots from the IGV viewer displaying RNA-seq read distributions encompassing *GAS5* in controls and *ZCCHC8* patients (E) Genome browser read coverage plots from the IGV viewer displaying RNA-seq read distributions encompassing *snoRD* genes in *GAS5* locus. (F) Meta-plot showing full-length (>/=5 kb) Long Interspersed Nuclear Element (*LINE1* or *L1*) expression in Controls (blue) and *ZCCHC8* cases (salmon) with standard error compared across scaled length of 5.5 kb and 1 kb flanks. (G–I) QPCR analysis of *GAS5*, *L1ORF1* and *L1ORF2* RNA transcripts from patient RNA samples of different DC genetic subtypes. Data represent means ± standard deviation, *n* = 3, with *P* values determined by one-way ANOVA were reported on graph (J) Gene ontology analysis for dysregulated gene signatures reveal genes involved in myeloid activation, ribosome biogenesis non-coding RNA processing and DNA metabolic process. (K–M) Dysregulated gene signatures and upregulation of inflammatory signalling pathways in ZCCHC8 patients as analysed by metascape tool (FDR < 0.05). The size of the circle correlates to number of genes involved in the process. Source data are available online for this figure.

characteristic of X-linked DC, typically resulting from variants in the *DKC1* gene (Heiss et al, 1998), our findings, in conjunction with reported germline variants in *POLA1* elsewhere contribute to diverse X-linked manifestations that include skin pigmentation, intellectual disability, hypogonadism, immune deficiency and bone marrow failure (Table 2).

POLA1-primase interacts with CST complex emphasizing its critical role in telomere protection and telomere replication (Chen et al, 2013; Diotti et al, 2015). Diminished catalytic function of variant POLA1 and its impact on binding to primase (PRIM1 and PRIM2A) and the CST complex can lead to loss of genome-wide DNA replication integrity in *POLA1* patients (Fig. 4). In line with this, biallelic *PRIM1* variants that affect DNA replication fork stability are reported in patients with primordial dwarfism with episodic thrombocytopenia and anaemia (Parry et al, 2020). Moreover, molecular characterisation of CTC1 variants revealed impaired interaction with POLA1-primase complex creating single-stranded gaps in the telomeric DNA (Chen et al, 2013). This discovery of novel *POLA1* and *POT1* variants (this study), in addition to the previously documented pathogenic variants in CST and shelterin component genes in other cases (Takai et al, 2016; Kelich et al, 2022; Walne et al, 2013; Simon et al, 2016; Polvi et al, 2012), who did not consistently display reduced leucocyte telomere length suggests *POLA1* and *POT1* pathogenic variants can cause DC or DCL features (Dataset EV1; Table 2) associated with variable telomere lengths. *POT1* variants reported here, appear to functionally mimic the ssDNA telomere binding defects (Fig. 3G,H), similar to *POT1* variants described in familial cancer patients without DC or DCL features (DeBoy et al, 2023; DeBoy et al, 2024; Kelich et al, 2022; Robles-Espinoza et al, 2014; Shi et al, 2014). The strong interactions between 53BP1 and TRF2, along with RPA70 accumulation and activation of ATR-CHK1-P53 signalling axis is a hallmark of POT1 deficiency (Wei et al, 2021) signifying that dysfunctional telomeres cause genome instability in these *POT1* patient cells (Fig. 4). Structural cryo-EM studies revealing the physical associations of CST with POLA1-primase (He et al, 2022; Cai et al, 2024) and POT1-TPP1 (Cai et al, 2024) complexes suggest the importance of these interactions in efficient telomere replication. In the context of pathogenic variants identified within *POLA1* and *POT1* that disrupted these interactions with the CST complex, can explain the pathogenetic basis of DC and DCL phenotypes observed in these patients.

Furthermore, we identified novel or rare pathogenic variants within *ZCCHC8*, that include a recurrent variant (p.Gly170Arg), in an allelic series of cases presenting DC or DCL features including an MDS phenotype (DCL 61; Fig. 5A; Table 4), which is a recognised complication among patients with inherited bone marrow failure. Intriguingly, previous studies linked ZCCHC8 haploinsufficiency to decreased *TERC* levels in human and murine cells (Gable et al, 2019). However, our findings suggest only minimal *TERC* processing defects in patients with ZCCHC8 deficiency (Fig. 5). Noting the lack of significant impact on *TERC* levels in patients' blood cells (Fig. 5), we focussed our analysis on other RNA species as prior research has highlighted the significance of ZCCHC8 in degradation of transcripts particularly originating from open chromatin regions (Garland et al, 2022). This analysis revealed dysregulation of snoRNAs encoding within lncRNA *GAS5* and *LITE* RNAs, suggesting a broader impact of *ZCCHC8* variants on nuclear RNA homoeostasis (Fig. 6). Nuclear *GAS5* acts as a decoy glucocorticoid response element sequestering glucocorticoid receptor response by binding to its DNA-binding domain. Furthermore, *GAS5* upregulation is shown to inhibit telomerase activity in cancer cells and decreases proliferative potential of cord blood CD34+ cells compromising haematopoietic colony formation in vitro (Wei et al, 2021; Kino et al, 2010; Du et al, 2024). Our observations in *ZCCHC8* patient blood cells also parallel previous findings in DC, where patient cells harbouring *DKC1* variants affect other snoRNAs beyond *TERC* (Nagpal et al, 2022) and upregulate the inflammatory response (Westin et al, 2023). Given the RNA dysregulation in *ZCCHC8* patients, the recent evolution of PAPD5 inhibitors that can restore functional RNA species in cell models of both DC (Boyraz et al, 2016; Shukla et al, 2020; Fok et al, 2019; Nagpal et al, 2020) and DCL disorder-(Poikiloderma with neutropenia-USB1 deficiency; Jeong et al, 2023) could hold some promise for restoring haematopoietic potential in these patients. It is also noteworthy that the glucocorticoid receptor agonists dexamethasone and prednisolone, exhibit positive effects in several DC and DCL patients. Hence, it is plausible to consider that specific targeting of *GAS5* regulation could present a new therapeutic approach for these patients.

In summary, our comprehensive analysis of a large series of DC and DCL families revealed the complex genetic architecture broadening our understanding of the molecular mechanisms, particularly in the context of ncRNA dysregulation that occur beyond *TERC*.

# Methods

### Reagents and tools table

| Reagent/resource | Reference or source | Identifier or catalogue number |
|---|---|---|
| **Experimental models** | | |
| HeLa-TIR1 | Gockert et al, 2022 | N/A |
| HeLa-ZCCHC8-3F-mAID | Gockert et al, 2022 | N/A |
| POT1 patient's cells | This study | N/A |
| 293T cells | ATCC | CRL-3216 |
| **Recombinant DNA** | | |
| POLA1 p.Arg616Gly pcDNA3.1(+)-C-eGFP | Genscript; This study | U1499ED100-2 |
| POLA1 p.Leu537Iso pcDNA3.1(+)-C-eGFP | Genscript; This study | U904MED290-1 |
| POLA1 p.Tyr1146Cys pcDNA3.1(+)-C-eGFP | Genscript; This study | U904MED290-3 |
| POLA1 p.Pro496S pcDNA3.1(+)-C-eGFP | Genscript; This study | U1499ED100-1 |
| STN1 pcDNA3.1(+)-N-DYK | Genscript; This study | U163LED050-1 |
| CTC1 pcDNA3.1(+)-N-6His | Genscript; This study | U163LED050-2 |
| POLA1 WT pcDNA3.1(+)-N-eGFP | Genscript; This study | U163LED050-4 |
| TEN1 pcDNA3.1(+)-N-HA | Genscript; This study | U163LED050-5 |
| POT1_WT_pcDNA3.1(+)-N-6XMyc | Genscript; This study | U2981HJ120-4 |
| POT1p.Arg432*STOP_pcDNA3.1(+)-N-6XMyc | Genscript; This study | U2981HJ120-6 |
| POT1p.Tyr419His_pcDNA3.1(+)-N-6XMyc | Genscript; This study | U2981HJ120-8 |
| POT1p.Pro146Leu_pcDNA3.1(+)-N-6XMyc | Genscript; This study | U2981HJ120-12 |
| POT1p.Gly551Arg_pcDNA3.1(+)-N-6XMyc | Genscript; This study | U2981HJ120-14 |
| POT1p.Cys503Gly_pcDNA3.1(+)-N-6XMyc | Genscript; This study | U2981HJ120-16 |
| POT1p.Ser243Asn_pcDNA3.1(+)-N-6XMyc | Genscript; This study | U2981HJ120-10 |
| ZCCHC8_OHu08841C_pcDNA3.1(+)-N-eGFP | Genscript; This study | U348VHH250-2 |
| ZCCHC8_OHu08841C_p.Glu66Lys_pcDNA3.1(+)-N-eGFP | Genscript; This study | U348VHH250-14 |
| ZCCHC8_OHu08841C_p.Glu113Lys_pcDNA3.1(+)-N-eGFP | Genscript; This study | U348VHH250-12 |
| ZCCHC8_OHu08841C_p.Gly170Arg_pcDNA3.1(+)-N-eGFP | Genscript; This study | U348VHH250-10 |
| ZCCHC8_OHu08841C_p.Gly184Glu_pcDNA3.1(+)-N-eGFP | Genscript; This study | U348VHH250-4 |
| ZCCHC8_OHu08841C_p.Gly196Lys_pcDNA3.1(+)-N-eGFP | Genscript; This study | U348VHH250-8 |
| ZCCHC8_OHu08841C_p.Val220Leu_pcDNA3.1(+)-N-eGFP | Genscript; This study | U348VHH250-6 |
| ZCCHC8_OHu08841C_pCMV-3Tag-4a | Genscript; This study | U348VHH250-26 |
| ZCCHC8_OHu08841C_p.Glu66Lys_pCMV-3Tag-4a | Genscript; This study | U348VHH250-30 |
| ZCCHC8_OHu08841C_p.Glu113Lys_pCMV-3Tag-4a | Genscript; This study | U348VHH250-32 |
| ZCCHC8_OHu08841C_p.Gly170Arg_pCMV-3Tag-4a | Genscript; This study | U348VHH250-34 |
| ZCCHC8_OHu08841C_p.Gly184Glu_pCMV-3Tag-4a | Genscript; This study | U348VHH250-36 |
| ZCCHC8_OHu08841C_p.Gly196Lys_pCMV-3Tag-4a | Genscript; This study | U348VHH250-28 |
| ZCCHC8_OHu08841C_p.Val220Leu_pCMV-3Tag-4a | Genscript; This study | U348VHH250-38 |
| 3X HA RBM7_MTREX_OHu08849C_pCMV-3Tag-1a-P2A | Genscript; This study | U348VHH250-42 |
| **Antibodies (dilutions)** | **Company** | **Catalogue no.** |
| POT1 | Protein Tech | 10581-1-AP |
| TPP1 | Cell Signal | 14667 |
| Phospho ATR | Abcam | Ab103970 |
| Phospho CHK1 | Abcam | Ab283261 |
| Phospho P53 | Cell Signal | 12571 |
| RPA70 | Santa Cruz | Sc-48425 |
| α-Tubulin | Abcam | Ab7291 |
| GFP | Santa Cruz | Sc-9996 |
| PRIM2A | Sigma-Aldrich | HPA046566 |
| PRIM1 | Sigma-Aldrich | SAB2501877 |
| Histidine | Cell Signal | 2365 |
| FLAG tag | Sigma-Aldrich | F-1804 |
| HA-tag | Protein Tech | 51064-2-AP |
| ZCCHC8 | Abcam | Ab68739 |
| MYC-tag | Protein Tech | 16286-1-AP |
| 53BP1 | Abcam | Ab175933 |
| TRF2 | Abcam | Ab 108997 |
| **Oligonucleotides and other sequence-based reagents** | | |
| PCR primers | This study | Appendix Table S6 |
| **Chemicals, enzymes and other reagents** | **Company** | **Catalogue no.** |
| Taqman fast advanced | ThermoFisher | 4444557 |
| Powerup Sybrgreen | ThermoFisher | A25742 |
| DNA-PK inhibitor NU7026 | STEMCELL Technologies | 74172 |
| ATM inhibitor KU55933 | Biotechne TOCRIS | 3544 |
| Auxin | Cambridge Bioscience | HY-134653 |
| DMEM high glucose GlutaMax | ThermoFisher | 10569010 |
| RPMI1640 GlutaMAX | ThermoFisher | 61870036 |
| MycoAlert® Mycoplasma detection kit | Lonza | LT07-701 |
| WesternBreeze Chemiluminescent Kit | ThermoFisher | WB7104, WB7106 |
| Duolink® In Situ detection kit | Sigma-Aldrich | DUO92002 |
| GFP-TRAP® Agarose | Chromtek | gta |
| TNT Quick Coupled Transcription/Translation System | Promega | L1170 |
| **Software** | | |
| GraphPad Prism | www.graphpad.com | N/A |
| Chimera | www.cgl.ucsf.edu/chimera/ | N/A |
| PROSize software | Agilent | N/A |
| GATK v3.2 | Broad Institute | N/A |

## Methods and protocols

### Patient samples

The patients included in this study had all been recruited to the London Dyskeratosis Congenita registry. Blood samples and clinical information were collected at enrolment. All participants and their family members provided written informed consent (London-City and East ref 07/Q0603/5) and all the experiments conducted using these samples conformed to the principles set out in the WMA Declaration of Helsinki and the Department of Health and Human Services Belmont Report.

### Variant calling and interpretation

Genomic DNA was extracted from peripheral blood samples (Puregene, Qiagen). Exome data from 418 samples was processed and called jointly with a set of 2500 WES internal control samples (UCL-ex consortium) using the recommendations from the Genome Analysis Toolkit (GATK v3.2) to minimize artefactual batch effects. The way of analysing the exome data depends partly on the perceived mode of inheritance. If it seems recessive, then we are looking for homozygous or biallelic variants in the same gene or genetic region. The heterozygote frequency of the variant needs to be less than 1/1000 so the overall calculated likelihood of occurrence is less than 1/1,000,000 (gnomAD database). For a dominant inheritance pattern, the maximum allowed frequency was 1/100,000.

### Telomere length analysis

High-throughput STELA was conducted following established procedures (Norris et al, 2021). In summary, we utilized 40 ng of genomic DNA (XpYp) or 20 ng (17p) in triplicate 30 μL PCR reactions. These reactions included telomere-adjacent primers designed for the XpYp telomere (XpYpC: 5′ -CAGGGACCGGGA-CAAATAGAC-3′) and the 17p telomere (17pseq1rev: 5′-GAATC-CACGGATTGCTTTGTGTAC-3′), as well as 1.5 U of a 10:1 mixture of Taq polymerase (ThermoFisher Scientific) and Pwo polymerase (Roche). The thermal cycling conditions comprised 23 cycles of 94 °C for 20 s, 65 °C for 30 s, and 68 °C for 5 min. The amplified fragments were separated using capillary gel electrophoresis, and the mean telomere length was determined through the utilization of PROSize software (Agilent).

### Cell culture plasmids and treatments

HeLa-TIR1 and HeLa-ZCCHC8-3F-mAID cells (kind gift from Torbeick Hensen; Aarhus Denmark) were cultured in Dulbecco's modified Eagle's medium (DMEM) and Epstein Barr virus-transformed lymphoblastoid lines (LCLs) were established from patients and unaffected individual blood samples and grown in RPMI1640. All culture media was supplemented with 10% (v/v) foetal bovine serum (FBS; HyClone),100 IU/ml penicillin, and 100 mg/ml streptomycin (Invitrogen). All cell lines were tested and authenticated to be free of mycoplasma contamination. All the plasmids used in the study were obtained as ready to use transfection quality cDNAs from Genscript. For endogenous ZCCHC8 depletion 750μM of indole-3 acetic acid (IAA) dissolved in DMSO was introduced into culture plates for required time points. ATM kinase inhibitor KU55933 was dissolved in DMSO and added to cell culture media at final concentration of 20μM. DNA-PK inhibitor NU7026 was dissolved in 300μM dimethyl fluoride and added to cell culture media at 50μM final concentration. PARN (sc-61297) and ZCCHC8 (sc-96061) small interference (si) RNAs were purchased from Santacruz and transfected using RNAimax (Invitrogen) into 293T cells cultured in complete DMEM and processed for analysis 48 h post transfection.

### RNA-seq analysis

Reads obtained from sequencing were mapped to hg38 genome with STAR aligner (Dobin et al, 2013) using options: "--peOver-lapNbasesMin 40 --peOverlapMMp 0.8 --genomeDir ./Genome_dir/ --readFilesIn R1_001.fastq.gz R2_001.fastq.gz --outFilterType BySJout --outFilterMultimapNmax 200 --alignSJoverhangMin 8 --alignSJDBoverhangMin 1 --outFilterMismatchNmax 999 --out-FilterMismatchNoverLmax 0.6 --alignIntronMin 20

--alignIntronMax 1000000 --limitOutSJcollapsed 5000000 --out-SAMattributes NH HI NM --outSAMtype BAM Unsorted --out-FileNamePrefix $base_RNA --outTmpDir $_temp --seedPerWindowNmax 10". The BAM files generated were either merged or used individually to create bigwig files utilizing the deeptools bamCoverage tool. Scaling factors, obtained through DESeq2 normalization, were applied to account for variations in read depth, and subsequent RPKM normalization was performed on these bigwig files. These RPKM-normalized bigwig files were employed for visualization in the Integrated Genome Viewer (IGV) and for generating plots. Counts for diverse genomic elements were acquired using the feature counts function from the subread package. These counts were derived for fragments and were subsequently utilized in the DESeq2 package for conducting differential analysis (Vulliamy et al, 2004). The data underwent transformation, specifically the rlog transformation, and were used to create heatmaps for the most significantly differentially expressed genes. Volcano plots were generated with the Enhanced Volcano package. Functional annotation of both upregulated and downregulated genes was conducted using the WEB-based GEne SeT AnaLysis Toolkit, which explored biological processes and disease pathways. To pinpoint enriched inflammatory pathways in ZCCHC8 cases, genes associated with inflammation, whether upregulated or downregulated, were subjected to analysis via the Metascape annotation tool (Zhou et al, 2019). For assessing the relative abundance of RNA-seq reads across various genomic elements, coverage of reads within these elements was computed. These elements encompassed PROMPTs (1 kb region upstream from the initial 100 bp up to the transcription start site), eRNAs for blood cell (PBMC)-specific enhancers (Guo et al, 2022), small and long ncRNAs, exons, introns, and TE (Transposable Element) regions as defined by Repeatmasker. The resulting counts were represented as pie charts and compared to determine the relative abundance between the two groups (Control and ZCCHC8 cases).

To perform a specific analysis of transposable elements (TEs), we initially obtained TE counts for various subfamilies using the TE transcripts tool (Love et al, 2014). These counts were then subjected to DESeq2 analysis to identify differential enrichment. At the individual locus level, we obtained counts for full-length LINE1 or L1 elements that exceeded 5 kilobases (Kb) in length. Next, we compared RNA-seq coverage by creating meta-plot profiles for LINE1 elements, plotting the signal from merged bigwigs between the control and ZCCHC8 cases using the Seqplots tool. To visualize the expression of full-length L1 elements at individual loci, we generated a circos plot comparing the RPKM (Reads Per Kilobase Million) levels of L1 elements on a chromosome ideogram. This was accomplished using the shinycircos webtool. For Long Terminal Repeat (LTR) subfamilies that included human endogenous retroviruses (HERVs) such as HERVK, HERVH, HERVL, ERVL, and ERV24, we compared their expression levels using the bigwigAverageOverBed tool. We then presented the results as boxed-violin plots with the ggplot2 package. Similar comparison plots were also created for non-coding RNA elements, including PROMPTs, eRNAs, and small ncRNAs. To identify upregulated small ncRNAs, we applied a statistical criterion (Wald test P value < 0.05) and determined the overlapping genes. These genes were further compared for changes in expression in the ZCCHC8 cases. To generate the boxed-violin plots, statistical analysis was performed using the two-sided Dunn test through the Rstatix

package, facilitating pairwise multiple comparisons of the ranked data. Gene pathway analysis is performed using Metascape tool (http://metascape.org; Zhou et al, 2019). All RNA-seq files were deposited in NCBI database under GEO accession codes: GSM7832028; GSM7832029; GSM7832030 for controls and GSM7832031; GSM7832032; GSM7832033 for *ZCCHC8* cases.

### Quantitative RT-PCR

RNA was extracted from whole blood patient cells. cDNA was prepared from total RNA using Invitrogen Superscript IV according to manufacturer's instructions using 600 ng input RNA from LCLs and 500 ng input RNA from blood and was primed with an equal mix of anchored dT oligonucleotides and random hexamers unless stated otherwise. TaqMan probes used were for detection of *ZCCHC8* and *PARN* amplicons. All reactions were set up using TaqMan™ Fast Advanced Master Mix according to the manufacturer's instructions. *TERC*, *LINE1* and *GAS5* expression levels were measured using gene-specific primers (Appendix Table S5). Reactions were set up using PowerUp Sybr Green Master mix. All samples were run in triplicate. Each GOI was normalised against the control gene(s) and fold change between cases and controls was calculated unless stated otherwise. TERRA levels are detected by the previously described qPCR method (Feretzaki and Lingner, 2017).

### DNA–RNA primer extension assays

Wild-type and variant forms GFP-POLA1 plasmids were expressed in 293T cells and immunoprecipitated with GFP-TRAP agarose beads (Chrometek). GFP-TRAP enriched proteomes were eluted in 0.2 M glycine and Tris buffered to pH 7.2 and subsequently used in the primer extension assay as previously described (Han et al, 2016). Briefly primer extension substrate was generated by combining 100 μL of a 5 μM RNA oligo with a fluorescein label, having a length of 15 nucleotides (5′-Fluorescein-rGrGrArArArGr-GrArCrGrArArArCrA-3′), 100 μL of a 7.5 μM DNA oligo spanning 25 to 40 nucleotides (5′-(A)nCCGGTGTTTCGTCCTTTCC-3′), and 100 μL of a 10× reaction buffer composed of 200 mM Tris-HCl (pH 7.8), 100 mM MgCl2, 20 mM DTT, and 500 mM NaCl. The mixture was heated to 75 °C and gradually allowed to cool to room temperature and confirmed by native PAGE for successful annealing. To test POLA1 activity, 1 μL of a threefold serially diluted CD437 toxin was preincubated with 10 μL of eluted POLA1 enriched proteome at room temperature for a duration of 15 min. Following this preincubation, the CD437-treated POLA1 enriched proteome was combined with 5 μL of the primer extension substrate, 5 μL of 10 mg/ml BSA, and 1 μL of 2.5 mM dNTP. The primer extension reaction was allowed to proceed at room temperature for 15 min and was then halted by adding 10 μL of gel loading buffer II (Life Technologies). The resulting reaction products were separated on a 15% TBE-UREA gels and visualized using iBright Imaging System.

### POT1 binding assays

To assess the functional implications of missense variants in POT1, we employed an in vitro translation reaction, utilizing the TNT Quick Coupled Transcription/Translation System from Promega, following the manufacturer's guidelines as described previously (Schratz et al, 2021). Briefly, a 50 μL reaction mixture, consisting of 1 μg of expression vector encoding POT1 wild type and variants,

20 μM methionine, and 40 μL of the TNT Quick Master mix with rabbit reticulocyte lysate, was incubated at 30 °C for 90 min and subjected to gel shift assay. The POT1 binding was determined in a 20 μL reaction mixture containing 2 μL of the in vitro translated product (IVT) combined with 10 nM IRDye 800-labelled, single-stranded telomeric oligonucleotide, 2.5 μM of non-specific single-stranded DNA (ssDNA), and 1 μg of poly(dI-dC) in a binding buffer (100 mM Tris, 500 mM KCl, 12.5 mM DTT, 10 mM EDTA and 0.25% Tween 20). We employed the telomere oligonucleotide 5′-GGTTAGGGTTAGGGTTAGGG-3′, and the non-specific ssDNA oligonucleotide 5′-TTAATTAACCCGGGGATCCGGCTT-GATCAACGAATGATCC-3′. The reaction mixtures were allowed to incubate for 30 min at room temperature, and subsequently, the protein-DNA complexes were separated by gel electrophoresis on a 10% polyacrylamide Tris-Borate EDTA gel, running at 100 V for 2 h at 4 °C. POT1 bound oligo products were visualised using the iBright imaging System, and quantified using ImageJ.

### Proximity ligation assay (PLA)

PLA was conducted utilizing the Duolink in situ kit (Sigma), following the manufacturer's guidelines. Cells were plated on poly lysine-coated slides fixed with Methanol Acetone or PFA, as previously outlined. After fixation, cells were incubated in Duolink blocking solution for 30 min at 37 °C. Primary antibodies (Appendix Table S6), were diluted in Duolink Antibody diluent and added to cells, which were incubated overnight at 4 °C. Subsequently, the cells were washed in Duolink Wash buffer A (2 × 5 min). PLA plus and minus probes were diluted 1:5 in antibody diluent and added to the cells, incubating for 1 h at 37 °C. Further washes in Duolink Wash buffer A (2 × 5 min) followed. The Ligation–Ligase mixture was prepared according to instructions and applied to cells for 30 min at 37 °C. After a wash (2 × 2 min in Duolink wash buffer A), the Duolink Amplification-Polymerase solution was added and incubated for 100 min at 37 °C. Subsequent washes in Duolink wash buffer B (2 × 10 min) were performed, and the cells were then mounted with Duolink in situ mounting medium with 4,6-diamidino-2-phenylindole before visualization using a Zeiss Confocal 710 microscope (Carl Zeiss). As a negative control, no primary antibodies were applied. Quantification was executed using Incell confocal microscopy (GE heath care) in DAPI stained region and analysed by Student's paired *t* test with GraphPad Prism (GraphPad, San Diego, USA).

### Co-immunoprecipitation (Co-IP) and western blotting

Western blotting was performed using primary antibodies (Appendix Table S7) and the corresponding alkaline phosphatase-conjugated secondary antibodies supplied in the WesternBreeze chemiluminescent kit (ThermoFisher). α-tubulin antibody was used us pull-down control against input. ATM inhibitor KU55933 (20 μM) and DNA-PK inhibitor NU7026 (50 μM) treated *POT1* patient cell protein extracts were prepared by lysing washed cells in denaturing buffer (9 M urea, 150 M 2-mercaptoethanol, 50 mM Tris-HCl pH 7.3) and subsequently sonicated to shear genomic DNA. Phosphorylated form of ATR was detected by separating proteins on 3–8% Tris glycine mini gels (Life Technologies). For all other proteins, 4–12% NuPAGE Bis-Tris mini gels (Life Technologies) were used. Gels were transferred onto PVDF membrane (GE Healthcare). Blotting was performed using specific primary antibodies (Appendix Table S6) and the corresponding alkaline

**The paper explained**

**Problem**

Dyskeratosis congenita (DC) is a rare inherited telomere biology disorder characterized by bone marrow failure. Despite genetic advances in DC, there are cases with DC and 'DC-like' (DCL) disease which remain uncharacterized at the genetic level.

**Results**

Through extensive genetic studies on a large cohort of clinically diagnosed DC and DCL cases, we identified several novel pathogenic variants within known genetic loci and discovered a new X-linked gene, *POLA1*. Functional curation of novel variants identified in *POLA1*, *POT1* and *ZCCHC8* genes expand the genetic and molecular basis associated with these conditions.

**Impact**

This study significantly advances our understanding of the current genetic landscape of DC and DCL disorders. By identifying novel pathogenic variants and elucidating their functional consequences, our work provides new insights into the mechanisms driving telomere maintenance and the inflammatory processes in DC and DCL disorders. These findings not only enhance the diagnostic capabilities for these rare conditions but also open up new avenues for targeted therapies.

phosphatase-conjugated secondary antibodies supplied in the WesternBreeze chemiluminescent kit (ThermoFisher). α-tubulin, β-actin and GAPDH antibodies were used as a loading control. For cell fractionation studies, HeLa cells were lysed in ice-cold HEPES containing 0.1% NP40, followed by centrifugation at 3000 rpm. The supernatant, containing cytoplasmic proteins, was collected. The pellet, which contained nuclei, underwent three brief washes in ice-cold HEPES with 0.1% NP40. It was then added to the pellet in ice-cold radioimmunoprecipitation assay (RIPA) buffer, supplemented with a cocktail of protease and phosphatase inhibitors (Roche, cat. no. 04693116001). After sonication (twice for 10 s at 50% pulse) to release nuclear proteins, the mixture was gently shaken on ice for 15 min. The nuclear fraction protein supernatant was obtained by centrifugation at $14,000 \times g$ for 15 min. The fractionated lysates were validated using antibodies against the nuclear TATA-binding protein (Abcam, cat. no. EPR3826) through western blotting, as described above.

### Structural analysis of the ZCCHC8 variants in NEXT complex

Cryo-EM structures of NEXT complex (PDB id: 7Z4Y; Gerlach et al, 2022), telomerase H/ACA RNP (PDB id: 8OUF; Ghanim et al, 2024), Telomerase bound TPP1-POT1 (PDB id: 7QXS; Sekne et al, 2022) POLA1-CST bound complex (PDB id 8D0B; He et al, 2022) are obtained from Protein Data Bank (http://www.rcsb.org). The location of ZCCHC8 variants and all structures were visualised using Chimera visualised (https://www.cgl.ucsf.edu/chimera/; Pettersen et al, 2004).

### Statistical analysis

Statistical analysis was performed with GraphPad Prism software (version 9), and a $P$ value < 0.05 was considered statistically significant. In bar graphs the overall significance was determined with one-way ANOVA with post hoc Mann–Whitney test was used to determine significant differences in between different groups. All experiments were conducted in non-blind manner and no case samples where available were excluded from the experiments.

## For more information

See, OMIM for dyskeratosis congenita: https://www.omim.org/entry/127550. Families of dyskeratosis congenita: www.dcaction.org. Telomere Network UK: https://genetics.org.uk/events/telomere-network-uk-ten/. Telomere biology disorder patient support and advocacy group: https://teamtelomere.org.

## Data availability

All novel *POLA1*, *POT1* and *ZCCHC8* variants are submitted to ClinVar database under accession numbers: VCV2626810.1, VCV2626808.1, VCV2626812.1, VCV2626811.1, VCV1029475.4, VCV2626809.1, VCV1917177.6, VCV2626817.1, VCV2626816.1, VCV541876.8, VCV850461.12, VCV2626815.1, VCV436390.21, VCV2626814.1, VCV2626807.1, VCV2626813.1. RNA-seq datasets were deposited in NCBI Gene Omni bus accession database under GEO accession code: GSE244915. Individual sample datasets can be accessed using. GSM7832028, GSM7832029, GSM7832030, GSM7832031, GSM7832032, GSM7832033.

The source data of this paper are collected in the following database record: biostudies:S-SCDT-10_1038-S44321-024-00118-x.

## Peer review information

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

## Acknowledgements

The authors would like to thank all the affected probands and their families for their participation in the study. The authors acknowledge financial support provided by UKRI/MRC (MR/P018440/1) and Blood Cancer UK (14032). The authors would like to thank Torben Heick Jensen from Aarhus University for kind donation of HeLa-TIR1 and HeLa-ZCCHC8-3F-mAID cells.

## Author contributions

**Hemanth Tummala**: Conceptualization; Resources; Data curation; Formal analysis; Supervision; Funding acquisition; Investigation; Methodology; Writing—original draft; Project administration; Writing—review and editing. **Amanda J Walne**: Formal analysis; Validation; Investigation; Methodology. **Mohsin Badat**: Formal analysis; Writing—review and editing. **Manthan Patel**: Data curation; Formal analysis; Validation; Investigation. **Abigail M Walne**: Formal analysis; Investigation; Methodology. **Jenna Alnajar**: Investigation; Methodology. **Chi Ching Chow**: Validation; Investigation; Methodology. **Ibtehal Albursan**: Investigation; Methodology. **Jennifer M Frost**: Investigation; Methodology. **David Ballard**: Methodology. **Sally Killick**: Investigation. **Peter Szitányi**: Investigation. **Anne M Kelly**: Investigation. **Manoj Raghavan**: Investigation. **Corrina Powell**: Investigation. **Reinier Raymakers**: Investigation. **Tony Todd**: Investigation. **Elpis Mantadakis**: Investigation. **Sophia Polychronopoulou**: Investigation. **Nikolas Pontikos**: Data curation; Formal analysis; Investigation; Visualization; Methodology. **Tianyi Liao**: Investigation. **Pradeep Madapura**: Investigation. **Upal Hossain**: Investigation. **Tom Vulliamy**: Investigation. **Inderjeet Dokal**: Conceptualization; Supervision; Funding acquisition; Investigation; Writing—review and editing.

Source data underlying figure panels in this paper may have individual authorship assigned. Where available, figure panel/source data authorship is listed in the following database record: biostudies:S-SCDT-10_1038-S44321-024-00118-x.

## Disclosure and competing interests statement

The authors declare no competing interests.

# Expanded View Figures

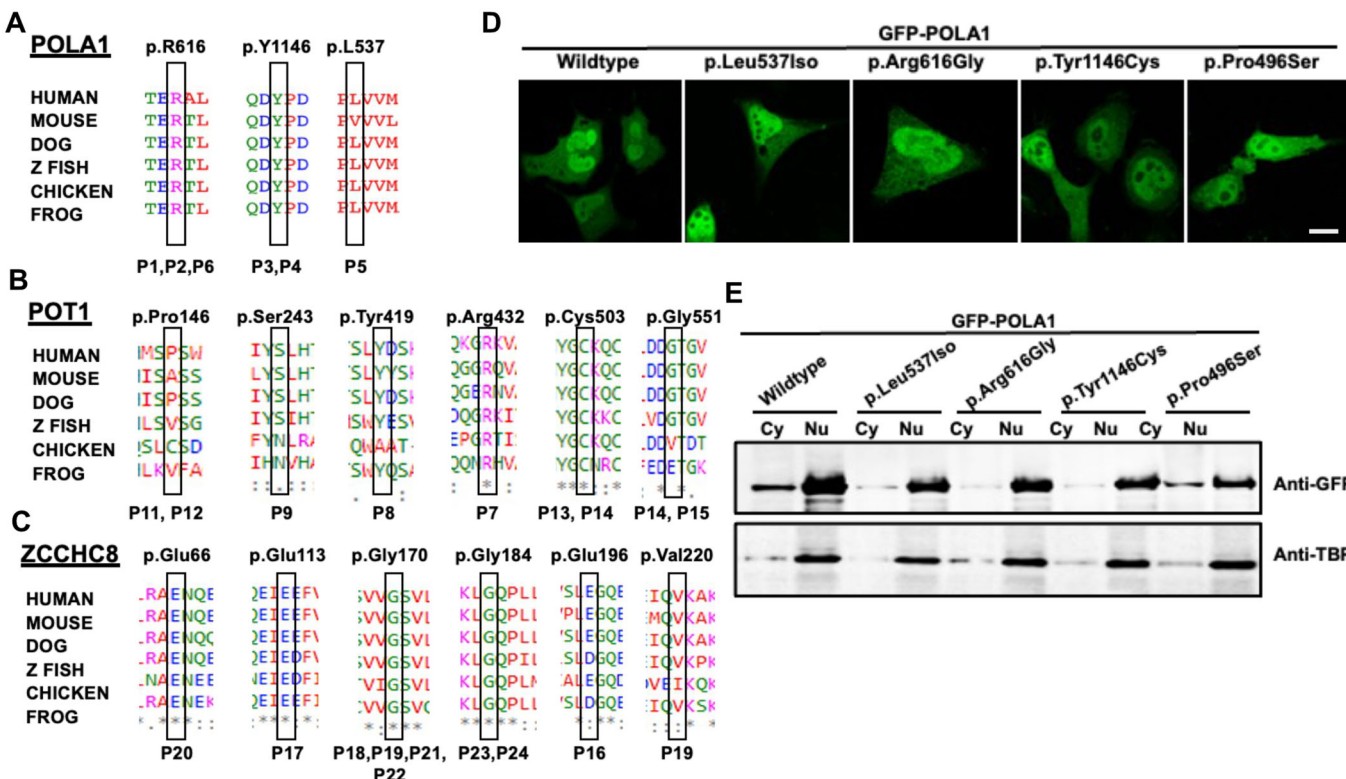

**Figure EV1. Germline variants in POLA1, POT1 and ZCCHC8.**

(A–C) Alignments were generated using Clustal omega, comparing the amino acid sequences around the sites of the variants identified in this study to other species. The high degree of conservation is evident, and the colour scheme indicates individual amino acid residue type assigned on basis of their profile by default parameters in Clustal omega. '*' indicates highly conserved ':' indicates semi conserved. (D) In HeLa cells expressing, confocal imaging of GFP-tagged POLA1 revealed predominantly nuclear and some cytoplasmic expression of POLA1. Scale bar, 20 μm. (E) This localisation is also confirmed by nuclear and cytoplasmic cell fractionation and subsequent western blotting. TATA-binding protein (TBP) antibody is used nuclear loading control. Source data are available online for this figure.

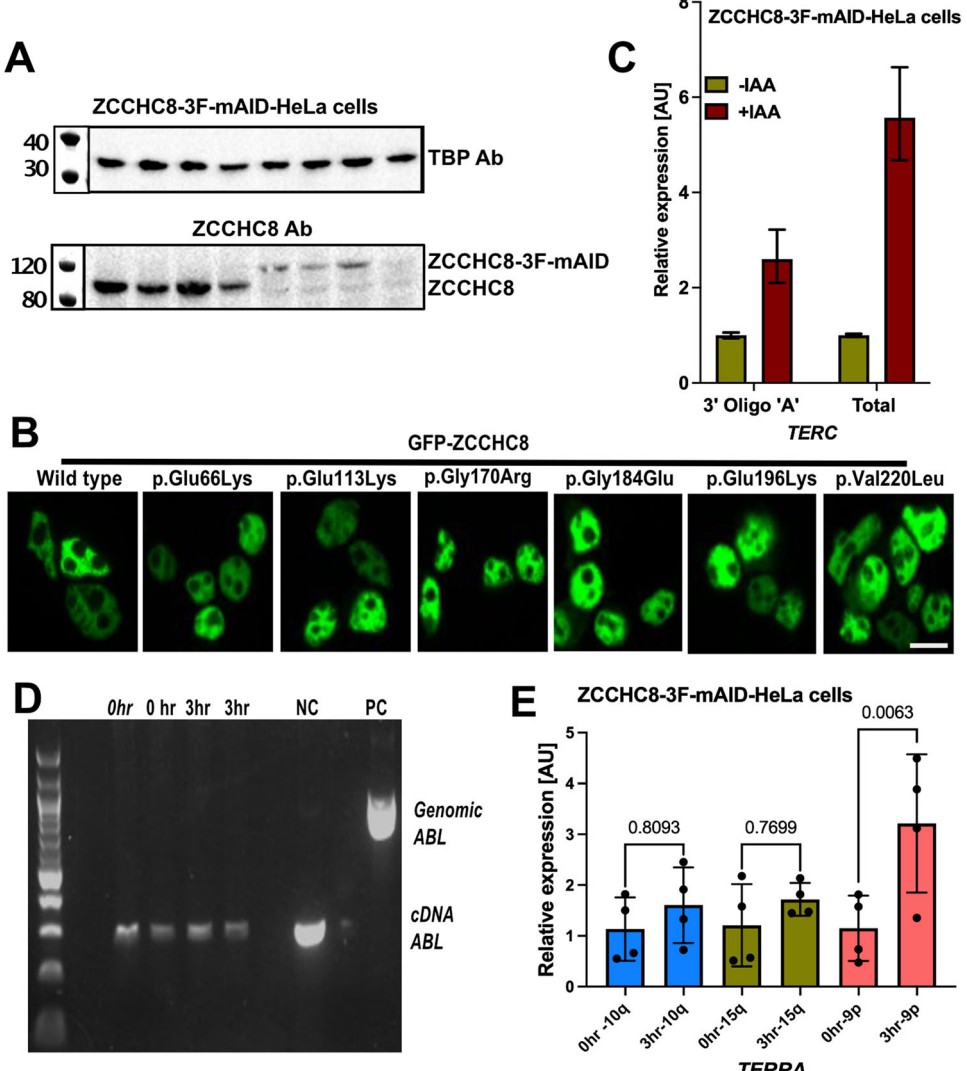

**Figure EV2. ZCCHC8 depletion increases *TERC* and *TERRA* transcripts in HeLa cells.**

(A) Immunoblot showing the reduction of ZCCHC8 protein levels after treatment with indole-3-acetic acid (IAA) at 750 μM for the indicated time points in HeLa-Tir1 cells and ZCCHC8-3F-mAID HeLa cells. TATA-binding protein (TBP) is used as a loading control. (B) Confocal images of GFP-tagged ZCCHC8 in HeLa cells. Panels are representative of images taken from different fields of view in three separate experiments. Scale bar, 20 μm. (C) Oligo-dT$_{(20)}$-primed mature (3'Oligo 'A') *TERC* RNA transcripts are distinguished from random hexamer-primed (Total) cDNA obtained from RNA samples from both untreated and treated ZCCHC8-3F-mAID HeLa cells with IAA. Data represent standard deviation calculated from means from upper and lower limits derived from $n = 2$ experiments run in triplicates for each condition. (D) Confirmation of RNA samples devoid of genomic DNA contamination as revealed by ABL primed transcript from cDNAs derived from untreated (0 h) and IAA treated (3 h and 24 h) ZCCHC8-3F-mAID HeLa cells. PC indicates genomic positive control. Data represent means ± standard deviation, from $n = 2$ experiments, with $P$ values determined by one-way ANOVA as reported on the graph. (E) Telomeric repeat containing RNA transcripts (*TERRA*) transcripts at indicated chromosomal locations were detected in cDNA samples from untreated (0 h) and IAA treated ZCCHC8-3F-mAID HeLa cells. Data represent means ± standard deviation, from $n = 2$ experiments, with p values determined by one-way ANOVA as reported on the graph. Source data are available online for this figure.

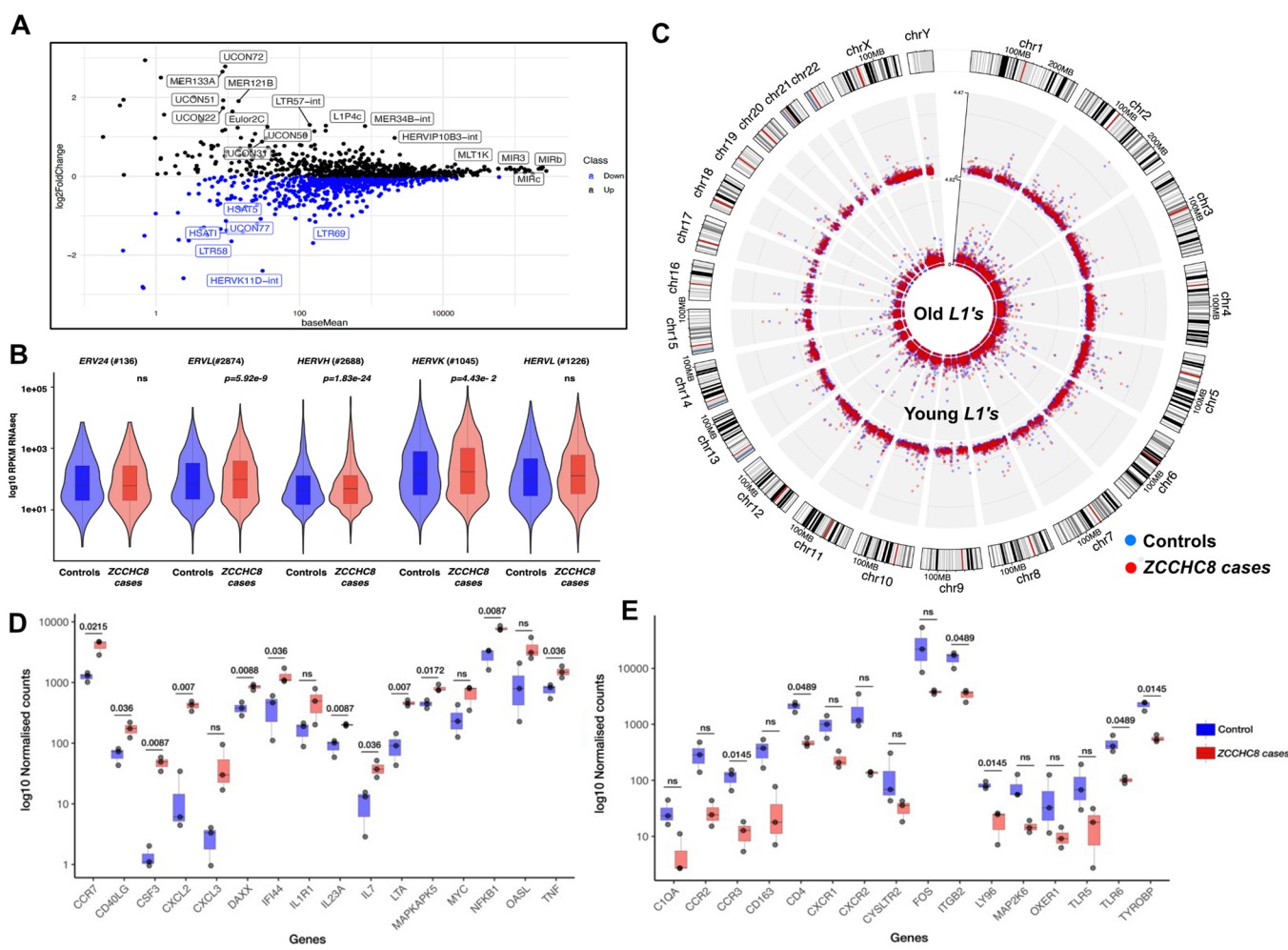

**Figure EV3. Transposable elements (TE's) dysregulation in ZCCHC8 patient blood.**

(A) Differentially expressed TE subfamilies (log2-fold change > 1 and the P-adjusted value < 0.05 after Benjamini–Hochberg multiple testing correction of Wald test P value of *ZCCHC8 cases* vs Controls; log2-fold change on y-axis and mean normalized counts on x-axis) showing either upregulated (black) or downregulated (blue). (B) Plots compare the expression of LTR (Long terminal repeats) subfamilies *ERV24, ERVL, HERVH, HERVK* and *HERVL* between Controls ($n = 3$; blue) and *ZCCHC8 cases* ($n = 3$; salmon). *P* values for all the violin/box plots were calculated using the pairwise two-sided multi-comparison Dunn test, a post hoc test, following Kruskal–Wallis test with Bonferroni correction. Violin-box plots indicate the median, bounds indicate the 25th and 75th percentiles, and whiskers limit show 1.5× interquartile range (C) Circos plot depicting expression of full-length L1 across chromosomal ideogram of younger L1s (*L1HS, L1PA2, L1PA3* and *L1PA4*) and older L1s (*L1PA5*-L1PA16, *L1M1-L1M4, L1P1-L1P4*) as RPKM levels for Controls ($n = 3$; blue) and *ZCCHC8 cases* ($n = 3$; salmon). (D, E) Box plot represents differentially expressed (FDR < 0.05) inflammatory responsive genes across three independent samples of controls ($n = 3$) and *ZCCHC8 cases* ($n = 3$). X-axis represent genes and Y-axis represents log10 normalised count. Violin-box plots indicate the median, bounds indicate the 25th and 75th percentiles, and whiskers limit show 1.5× interquartile range.

