## [Peer Review File · EMBO Molecular Medicine]

The evolving genetic landscape of telomere biology disorder dyskeratosis congenita

Hemanth Tummala, Amanda Walne, Mohsin Badat, Abigail Walne, Jenna Alnajar, Chiching Chow, Ibtehal Albursan, Jennifer Frost, David Ballard, Sally Killick, Peter Sztányi, Anne Kelly, Manoj Raghavan, Corrina Powell, Reinier Raymakers, Tony Todd, Elpis Mantadakis, Sophia Polychronopoulou, Nikolas Pontikos, Tianyi Liao, Manthan Patel, Upal Hossain, Tom Vulliamy, Inderjeet Dokal, and Pradeep Madapura

Corresponding author: Hemanth Tummala (h.tummala@qmul.ac.uk)

Review Timeline:

Submission Date:	12th Apr 24
Editorial Decision:	15th May 24
Revision Received:	7th Jun 24
Editorial Decision:	1st Jul 24
Revision Received:	8th Jul 24
Editorial Decision:	10th Jul 24
Revision Received:	12th Jul 24
Accepted:	18th Jul 24

Editor: Poonam Bheda

Transaction Report:

15th May 2024

Dear Dr. Tummala,

Thank you for the submission of your manuscript to EMBO Molecular Medicine. We have now received feedback from the three reviewers who agreed to evaluate your manuscript. As you will see from the reports below, the referees acknowledge the interest of the study and are overall supporting publication of your work pending appropriate revisions.

Addressing the reviewers' concerns in full in a point-by-point response will be necessary for further considering the manuscript in our journal, and acceptance of the manuscript will entail a second round of review. EMBO Molecular Medicine encourages a single round of revision only and therefore, acceptance or rejection of the manuscript will depend on the completeness of your responses included in the next, final version of the manuscript. For this reason, and to save you from any frustrations in the end, I would strongly advise against returning an incomplete revision. If you would like to discuss further the points raised by the referees, I am available to do so via email or video. Let me know if you are interested in this option.

We are expecting your revised manuscript within three months, if you anticipate any delay, please contact us. When submitting your revised manuscript, please carefully review the instructions that follow below. We perform an initial quality control of all revised manuscripts before re-review; failure to include requested items will delay the evaluation of your revision.

We require:

4) A .docx formatted letter INCLUDING the reviewers' reports and your detailed point-by-point responses to their comments. As part of the EMBO Press transparent editorial process, the point-by-point response is part of the Review Process File (RPF), which will be published alongside your paper.

5) A complete author checklist, which you can download from our author guidelines (<https://www.embopress.org/page/journal/17574684/authorguide#submissionofrevisions>). Please insert information in the checklist that is also reflected in the manuscript. The completed author checklist will also be part of the RPF.

6) Please note that all corresponding authors are required to supply an ORCID ID for their name upon submission of a revised manuscript.

7) It is mandatory to include a 'Data Availability' section after the Materials and Methods. Before submitting your revision, primary datasets produced in this study need to be deposited in an appropriate public database, and the accession numbers and database listed under 'Data Availability'. Please remember to provide a reviewer password if the datasets are not yet public (see <https://www.embopress.org/page/journal/17574684/authorguide#dataavailability>).

In case you have no data that requires deposition in a public database, please state so in this section. Note that the Data Availability Section is restricted to new primary data that are part of this study. This study includes no data deposited in external repositories.

8) For data quantification: please specify the name of the statistical test used to generate error bars and P values, the number (n) of independent experiments (specify technical or biological replicates) underlying each data point and the test used to calculate p-values in each figure legend. The figure legends should contain a basic description of n, P and the test applied. Graphs must include a description of the bars and the error bars (s.d., s.e.m.). Please provide exact p values.

9) Our journal encourages inclusion of *data citations in the reference list* to directly cite datasets that were re-used and obtained from public databases. Data citations in the article text are distinct from normal bibliographical citations and should directly link to the database records from which the data can be accessed. In the main text, data citations are formatted as follows: "Data ref: Smith et al, 2001" or "Data ref: NCBI Sequence Read Archive PRJNA342805, 2017". In the Reference list, data citations must be labeled with "[DATASET]". A data reference must provide the database name, accession number/identifiers and a resolvable link to the landing page from which the data can be accessed at the end of the reference.

Further instructions are available at .

- the medical issue you are addressing,

- the results obtained and

- their clinical impact.

13) Author contributions: CRediT has replaced the traditional author contributions section because it offers a systematic machine readable author contributions format that allows for more effective research assessment. Please remove the Authors Contributions from the manuscript and use the free text boxes beneath each contributing author's name in our system to add specific details on the author's contribution. More information is available in our guide to authors.

Please also suggest a visual abstract to illustrate your article as a jpeg file 550 px wide x 300-600 px high.

Share synopsis text and image, as well as eTOC:

Please note that these would be the final versions and changes during proofing are usually not allowed

16) As part of the EMBO Publications transparent editorial process initiative (see our policy here:

https://www.embopress.org/transparent-process#Review_Process), EMBO Molecular Medicine will publish online a Peer Review File (PRF) to accompany accepted manuscripts.

In the event of acceptance, this file will be published in conjunction with your paper and will include the anonymous referee reports, your point-by-point response and all pertinent correspondence relating to the manuscript. Let us know whether you agree with the publication of the PRF and as here, if you want to remove or not any figures from it prior to publication.

I look forward to receiving your revised manuscript.

Yours sincerely,

Poonam Bheda

Poonam Bheda, PhD
Scientific Editor
EMBO Molecular Medicine

***** Reviewer's comments *****

Referee #1 (Comments on Novelty/Model System for Author):

The medical impact is direct in terms of diagnosis of a rare disease, not treatment.

Referee #1 (Remarks for Author):

The authors of this work initially searched the genetic variants found in a large cohort of Dyskeratosis congenita (DC) and dyskeratosis congenita-like (DKCL) patients. They found variants in genes previously described in DC patients including several new ones. Most of these genes were also mutated in DCL patients. In addition, they found variants in a gene not described previously in DC patients, POLA1. Several new variants were also found in two genes described only in a few cases previously, POT1 and ZCCHC8. The possible functional consequences of the mutations found in these three genes were approached in the rest of the article. In summary, the authors describe that POLA1 variants showed decrease catalytic efficiency. POT1 variants showed decreased binding to single strand telomeric DNA. In addition, both POLA1 and POT1 variants impaired the interaction of these proteins with CST and the shelterin complex, which can affect telomere elongation. The analysis of ZCCHC8 variants indicated a general alteration of transcription regulation, including non-coding RNAs and repetitive elements and also other genes that might trigger inflammation.

These results represent a significant contribution to the genetic study of DC and DCL diseases reporting a new gene involved and unrevealing new protein interactions and cellular pathways that are altered in these diseases. The study is very extensive and the data presented sound, supporting the conclusions of the authors.

I would like comment some points as follows:

1. I have some concern about the results of the electrophoretic mobility shift assay shown in figure 3G. The blot shown seems to indicate that there is some DNA binding for the variants p.Pro146Leu and p.Ser243Asn. At least as observed for the higher specific band. However, the figure shown in panel 3H indicates very little or no binding for these variants. What could be the reason for this apparent discrepancy?
2. The authors analyze POT1 expression in patient cells in Figure 4D. They observe the presence of a prominent band of lower molecular weight than the wild type protein. Do the authors have any insight on the possible origin of this band? Have they performed RT-PCR experiments to determine if a cDNA of small size was expressed in these patients? If that were the case, sequencing of variant cDNAs could provide valuable information about this protein.
3. The possible role of the ZCCHC8 variants on TERC levels presented in this article indicate that there is no effect either in the mature or immature forms of TERC in whole blood RNA from these patients. These results seem to be contradictory to previous results showing that loss of ZCCHC8 function results in decreased TERC transcripts, as the authors indicate in the article. One possible difference between the studies is that the previous ones were made in cell lines while the present one is made in blood samples. Since the authors have generated HeLa-TIR1 cells expressing the different ZCCHC8 mutants, it would be of interest to know if these mutants induce decreases in TERC expression levels in the cells. This system might be more similar to the one used in previous studies.
4. There seems to be a contradiction in the family pedigree shown for patient DCL215 in Figure 5A since both parents are wild type and both patients heterozygous for the variant.
5. The bibliographic references are indicated by the authors' name in the manuscript text and by order of citation in the References section.

Referee #2 (Comments on Novelty/Model System for Author):

The authors have done an admirable job of assembling a large summary dataset for individuals with dyskeratosis congenita (DC) and DC-like (DCL) syndrome. Overall, the data are sound and advance the literature, especially with the discovery of a new gene, POLA1, as associated with DC and detailed studies of new variants in POT1 and ZCCHC8 in disease. However, it was difficult to read because of this complexity and I found it lacking sufficient detail in any one area.

Please note my expertise is much more on the clinical phenotypes than the mechanisms of these specific genes, and that is where I have focused my review.

Referee #2 (Remarks for Author):

The authors present a very ambitious study seeking to compile novel and known causes of DC and DCL by describing their large cohort and conducting mechanistic studies on a new putative DC gene, POLA1, and on new variants in POT1 and ZCCHC8. My comments focus predominately on the clinical and genetic aspects of the study.

Results section:

clarify whether the number of exomes is the same as the number of probands or affecteds, or family members. In the 461 families, how many were affected and how many sequenced (same for DCL). These data are buried in the Appendix.

The POLA1 data are compelling. Additional basic genomics, including frequencies of variants in publicly available databases, clinical outcomes, etc would improve the study.

The individuals with POT1 heterozygous variants are interesting given the connections between long telomeres and non-DC related cancers in that setting. Since the authors have several DC/DCL patients with just one POT1 variant, how do they reconcile the studies suggesting a different phenotype?

ZCCHC8 data suggest that TERC is not affected but instead that inflammation is a key component. It seems unlikely that inflammation "alone" would cause a DC/DLC phenotype, which is how I interpreted the data. What is the connection with telomere biology?

The clinical aspects could be strengthened by additional analyses of clinical outcomes.

Referee #3 (Remarks for Author):

The study submitted by Tummala et al. is a comprehensive study of the genetic landscape in dyskeratosis patients with data from one of the largest cohorts of patients studied to date.

The authors identify a novel gene associated with the disease (POLA1) and expand the number of patients associated with mutations in POT1 and ZCCHC8).

This is a very relevant study to expand the knowledge of this complex disease, and this reviewer believes that the publication of this paper should be prioritized.

Minor comments:

The number of patients with mutations identified in this study is small compared to the total number of patients with unclassified mutations. It would be interesting for the authors to comment on the possible reasons related to the difficulties in identifying the mutations in the remaining patients.

In the case of patients with mutations in the POLA1 gene, the reduction in telomere length is lower than in other DC associated genes, can the authors explain or hypothesize the reason for this difference? Could this indicate that patients with POLA1 mutations may have a milder phenotype?

Response to Editor

All additional changes have been highlighted in yellow in the manuscript.

2) Individual production quality figure files as .eps, .tif, .jpg (one file per figure). For guidance, download the 'Figure Guide PDF'

(<https://www.embopress.org/page/journal/17574684/authorguide#figureformat>).

We have now provided the source data consisting of Western blots and gel images in power point file and the analysis files in Excel format upon request from the source data coordinator.

4) A .docx formatted letter INCLUDING the reviewers' reports and your detailed point-by-point responses to their comments. As part of the EMBO Press transparent editorial process, the point-by-point response is part of the Review Process File (RPF), which will be published alongside your paper.

A point by point response has been included as docx file labelled **response to reviewers comments**.

5) A complete author checklist, which you can download from our author guidelines (<https://www.embopress.org/page/journal/17574684/authorguide#submissionofrevisions>). Please insert information in the checklist that is also reflected in the manuscript. The completed author checklist will also be part of the RPF.

6) Please note that all corresponding authors are required to supply an ORCID ID for their name upon submission of a revised manuscript.

ORCID ID of the corresponding author 'Hemanth Tummala' has been supplied.

7) It is mandatory to include a 'Data Availability' section after the Materials and Methods. Before submitting your revision, primary datasets produced in this study need to be deposited in an appropriate public database, and the accession numbers and database listed under 'Data Availability'. Please remember to provide a reviewer password if the datasets are not yet public

(see <https://www.embopress.org/page/journal/17574684/authorguide#dataavailability>).

We have now Included Data availability section in the manuscript.

This study includes no data deposited in external repositories.

8) For data quantification: please specify the name of the statistical test used to

generate error bars and P values, the number (n) of independent experiments (specify technical or biological replicates) underlying each data point and the test used to calculate p-values in each figure legend. The figure legends should contain a basic description of n, P and the test applied. Graphs must include a description of the bars and the error bars (s.d., s.e.m.). Please provide exact p values.

This has been addressed and explained in the figure legends where statistical analysis has been used to determine the significance P value as highlighted in the manuscript.

9) Our journal encourages inclusion of *data citations in the reference list* to directly cite datasets that were re-used and obtained from public databases. Data citations in the article text are distinct from normal bibliographical citations and should directly link to the database records from which the data can be accessed. In the main text, data citations are formatted as follows: "Data ref: Smith et al, 2001" or "Data ref: NCBI Sequence Read Archive PRJNA342805, 2017". In the Reference list, data citations must be labeled with "[DATASET]". A data reference must provide the database name, accession number/identifiers and a resolvable link to the landing page from which the data can be accessed at the end of the reference. Further instructions are available

at <https://www.embopress.org/page/journal/17574684/authorguide#referencesformat>

We have followed the suggested instructions.

<https://www.embopress.org/page/journal/17574684/authorguide#expandedview>

We have now included a new list of Tables 1-4, Table EV1-2 and new Figures EV1-3 as separate image files with the revised manuscript. The confocal image panels in Fig. 5E are included in Fig.EV2B.

This may be edited to ensure that readers understand the significance and context of

the research. Please refer to any of our published articles for an example.
We have written this piece emphasizing major findings for the non-specialist reader about the disease dyskeratosis congenita in the revised manuscript on page 2 as highlighted.

We have included some links to web resources on disease dyskeratosis congenita in the revised version of the manuscript.

13) Author contributions: CRediT has replaced the traditional author contributions section because it offers a systematic machine readable author contributions format that allows for more effective research assessment. Please remove the Authors Contributions from the manuscript and use the free text boxes beneath each contributing author's name in our system to add specific details on the author's contribution. More information is available in our guide to authors.

This section has been removed from the main manuscript document in the revised version.

There are no changes or declarations to be made in the competing interest section.

We have now included a Synopsis section summarising the key points.

Please also suggest a visual abstract to illustrate your article as a jpeg file 550 px wide x 300-600 px high.

Share synopsis text and image, as well as eTOC:

Please note that these would be the final versions and changes during proofing are usually not allowed

We have created a graphical representation of the genetic landscape of dyskeratosis congenita as a power point file .

Response to Reviewers

***** Reviewer's comments *****

Referee #1 (Comments on Novelty/Model System for Author):

The medical impact is direct in terms of diagnosis of a rare disease, not treatment.

Referee #1 (Remarks for Author):

The authors of this work initially searched the genetic variants found in a large cohort of Dyskeratosis congenita (DC) and dyskeratosis congenita-like (DKCL) patients. They found variants in genes previously described in DC patients including several new ones. Most of these genes were also mutated in DCL patients. In addition, they found variants in a gene not described previously in DC patients, POLA1. Several new variants were also found in two genes described only in a few cases previously, POT1 and ZCCHC8. The possible functional consequences of the mutations found in these three genes were approached in the rest of the article. In summary, the authors describe that POLA1 variants showed decrease catalytic efficiency. POT1 variants showed decreased binding to single strand telomeric DNA. In addition, both POLA1 and POT1 variants impaired the interaction of these proteins with CST and the shelterin complex, which can affect telomere elongation. The analysis of ZCCHC8 variants indicated a general alteration of transcription regulation, including non-coding RNAs and repetitive elements and also other genes that might trigger inflammation.

These results represent a significant contribution to the genetic study of DC and DCL diseases reporting a new gene involved and unrevealing new protein interactions and cellular pathways that are altered in these diseases. The study is very extensive and the data presented sound, supporting the conclusions of the authors.

We are grateful to this reviewer for these positive comments.

I would like comment some points as follows:

1. I have some concern about the results of the electrophoretic mobility shift assay shown in figure 3G. The blot shown seems to indicated that there is some DNA binding for the variants p.Pro146Leu and p.Ser243Asn. At least as observed for the higher specific band. However, the figure shown in panel 3H indicates very little or no binding for these variants. What could be the reason for this apparent discrepancy?

We appreciate the reviewer, for the careful analysis of the EMSA results presented in Figure 3G and H and the opportunity to clarify this apparent discrepancy. Our initial densitometry analysis just included the bottom specific band but not the higher molecular weight specific band. To address this concern, we have redone the analysis by including both specific bands and the result of this new analysis is presented in the Figure 3H, which now clearly shows that C-terminal POT1 mutants have less DNA binding activity due to absence of higher order DNA binding species as revealed by EMSA. We have clarified this in the figure legend on page 19.

2. The authors analyze POT1 expression in patient cells in Figure 4D. They observe the presence of a prominent band of lower molecular weight than the wild type protein. Do the authors have any insight on the possible origin of this band? Have they performed RT-PCR experiments to determine in a cDNA of small size was expressed in these patients? If that were the case, sequencing of variant cDNAs could provide valuable information about this protein.

We thank the reviewer for highlighting the observation regarding the lower molecular weight band in Figure 4D. We have performed cDNA sequencing in both blood and

EBV lymphoblastoid lines but failed to detect any abnormal short transcript. We believe this could be the result of post translational modification that is specifically occurring in POT1 patient cells as a consequence of the variant protein in these cells. To clarify this, we have included a statement in the relevant sections of page 5 as follows “Sequencing the cDNA from these patient cells did not reveal any mis-spliced transcript. This suggests that the observed lower molecular weight POT1 protein could be a direct result of an unknown post-translational modification occurring as consequence of POT1 variants.”

3. The possible role of the ZCCHC8 variants on TERC levels presented in this article indicate that there is no effect either in the mature or immature forms of TERC in whole blood RNA from these patients. These results seems to be contradictory to previous results showing that loss of ZCCHC8 function results in decreased TERC transcripts, as the authors indicate in the article. One possible difference between the studies is that the previous ones were made in cell lines while the present one is made in blood samples. Since the authors have generated HeLa-TIR1 cells expressing the different ZCCHC8 mutants, it would be of interest to know in these mutants induce decreases TERC expression levels in the cells. This system might be more similar to the one used in previous studies.

We appreciate the reviewer for pointing out the observed discrepancies in regard to the role of ZCCHC8 variants on TERC levels. While we acknowledge the differences in cell systems, we believe our current findings provide more comprehensive and robust role of ZCCHC8 variants in general RNA dysregulation rather just TERC. Specifically, we have consistently observed no or minimal change in both 3' and total TERC transcripts in RNAi treated 293 cells (Figure 5I) and Patient's blood (Figure 5J and K). Furthermore, in support of our findings, we have detected robust increase in 3' TERC species in the blood RNA of patients with biallelic variants in TERC specific deadenylase PARN. To address further as suggested by the reviewer, we have performed the experiments in the ZCCHC8-3F-mAID HeLa cells and the results are shown in the Fig. EV2. We have explained these findings in the manuscript on page 7 at the end of the second paragraph as below.

“This apparent discrepancy in TERC regulation between our study and those previously reported by Gable et al (2019), could be due to the complete absence of ZCCHC8 in their genetically knockout ZCCHC8^{-/-} HCT116 cell lines. In contrast, patient fibroblasts carrying the ZCCHC8 variant, as reported by Gable et al. (2019), show mild levels of ZCCHC8 protein expression, that is similar to our RNAi knockdown studies (Fig. 5H, I). However, acute depletion of ZCCHC8 with IAA treatment in ZCCHC8-3F-mAID HeLa cells, significantly increased both 3' polyadenylated and total forms of TERC, as well as Telomeric Repeat containing RNA-TERRA (Fig. EV2C, EV2E). Altogether, these results indicate that inherited ZCCHC8 deficiency has minimal impact on TERC regulation in the patient cells”.

We believe that acute depletion of ZCCHC8 by IAA treatment in short term may have global impact due to NEXT complex impairment that could lead to increase of several RNA transcripts with no specificity. The consistency of our findings across diverse cell systems suggests that ZCCHC8 deficiency may lead to general RNA dysregulation, rather than exerting a specific effect only on TERC. This new data has resulted in the inclusion of a new reference, Feretzaki et al 2017, in the references section of the revised manuscript.

4. There seems to be a contradiction in the family pedigree shown for patient DCL215 in Figure 5A since both parents are wild type and both patients heterozygous for the variant.

We thank the reviewer for pointing out the apparent contradiction in the family pedigree for the patient DCL215 in Figure 5A. One possible explanation for this could be that one of the affected parents might be a gonadal/germline mosaic for this ZCCHC8 variant. This has been previously reported in DC (Vulliamy et al 1999). A second possibility is that the “affected parent” might have undergone reversion of the mutant allele to wildtype; that is often referred to as reversion and has been reported in DC and other inherited bone marrow failure syndromes such as Fanconi anaemia. A third possibility is that there is non-paternity (i.e. the father is not the biological father). We have added the following statement as below in Page 6 third paragraph.

“Specifically, in DCL 215 family both siblings presented with this recurrent ZCCHC8 variant (p.Gly170Arg), while both parents when tested appear to be wildtype. The chances of both non-twin siblings developing a *de novo* disease-causing variant are unlikely. Therefore, we suspect that one of the parents, who carries the variant, is a germline or somatic mosaic, as such events have been previously observed in DC (Vulliamy et al., 1999).” We have included the new reference- Vulliamy et al., 1999 in the references section of the revised manuscript.

5. The bibliographic references are indicated by the authors' name in the manuscript text and by order of citation in the References section.

We thank the reviewer for spotting this, and the numbering has been removed from the references.

Referee #2 (Comments on Novelty/Model System for Author):

The authors have done an admirable job of assembling a large summary dataset for individuals with dyskeratosis congenita (DC) and DC-like (DCL) syndrome. Overall, the data are sound and advance the literature, especially with the discovery of a new gene, POLA1, as associated with DC and detailed studies of new variants in POT1 and ZCCHC8 in disease. However, it was difficult to read because of this complexity and I found it lacking sufficient detail in any one area.

Please note my expertise is much more on the clinical phenotypes than the mechanisms of these specific genes, and that is where I have focused my review.

We are grateful to this reviewer for acknowledging the significance of our study as our primary goal was to report a comprehensive resource that integrates clinical and genetic data to advance the understanding of DC and DCL disorders. While we aimed to provide a broad overview, we do recognize that the breadth and complexity of data can present with challenges to the reader with specific focus on clinical phenotypes. To address this, we have now provided expanded view tables (Tables 2-4), which give clinical features of patients with variants in *POLA1*, *POT1* and *ZCCHC8* genes.

Referee #2 (Remarks for Author):

The authors present a very ambitious study seeking to compile novel and known

causes of DC and DCL by describing their large cohort and conducting mechanistic studies on a new putative DC gene, *POLA1*, and on new variants in *POT1* and *ZCCHC8*. My comments focus predominately on the clinical and genetic aspects of the study.

Results section:

clarify whether the number of exomes is the same as the number of probands or affecteds, or family members. In the 461 families, how many were affected and how many sequenced (same for DCL). These data are buried in the Appendix.

The number of cases that had whole exome sequencing, targeted sequencing and whole genome sequencing are now clarified in the first paragraph of the results section on page 3.

The *POLA1* data are compelling. Additional basic genomics, including frequencies of variants in publicly available databases, clinical outcomes, etc would improve the study.

The variants identified in *POLA1* are all novel as indicated in Table 1 in the revised manuscript.

The individuals with *POT1* heterozygous variants are interesting given the connections between long telomeres and non-DC related cancers in that setting. Since the authors have several DC/DCL patients with just one *POT1* variant, how do they reconcile the studies suggesting a different phenotype?

It is well known that recurring variants in *DKC1* (p.Ala353Val), *TINF2* (p.Arg282His), *ACD* (p.Lys170del) as well as several recurring telomerase (*TERC* and *TERT*) variants cause mild to very severe phenotypes of DC. In light of these observations, it is not surprising that these particular variants in *POT1* (p.Pro146Leu), *POLA1* (p.Arg616Gly) and *ZCCHC8* (p.Gly170Arg) can cause both DC and DCL phenotypes.

ZCCHC8 data suggest that *TERC* is not affected but instead that inflammation is a key component. It seems unlikely that inflammation "alone" would cause a DC/DCL phenotype, which is how I interpreted the data. What is the connection with telomere biology?

We appreciate this critical comment by the reviewer. Noting the abnormal non-coding RNA dysregulation and high inflammation signature in patient blood cells we believe this could impact telomere maintenance in these patients. For example, elevation of *GAS5* non-coding RNA, is reported to inhibit *TERT* expression (ref: Wei X et al 2021) and disrupt haematopoiesis (ref: Du YX et al; 2024), which we refer to in the second paragraph of the discussion section on page 9 of the paper.

The clinical aspects could be strengthened by additional analyses of clinical outcomes.

Based on the available patient information provided by relevant clinicians, we have accurately described the clinical features of the patients in Table 2 for *POLA1* variants, Table 3 for *POT1* variants and Table 4 for *ZCCHC8* variants.

Referee #3 (Remarks for Author):

The study submitted by Tummala et al. is a comprehensive study of the genetic landscape in dyskeratosis patients with data from one of the largest cohorts of patients studied to date. The authors identify a novel gene associated with the disease (POLA1) and expand the number of patients associated with mutations in POT1 and ZCCHC8).

This is a very relevant study to expand the knowledge of this complex disease, and this reviewer believes that the publication of this paper should be prioritized.

We thank this reviewer for this very positive feedback and recognizing the significance of our study in advancing the understanding of DC. We also appreciate the endorsement of this reviewer for prioritizing the publication of our study.

Minor comments:

The number of patients with mutations identified in this study is small compared to the total number of patients with unclassified mutations. It would be interesting for the authors to comment on the possible reasons related to the difficulties in identifying the mutations in the remaining patients.

We thank the reviewer for the comment regarding the proportion of patients with identified mutations relative to those with unclassified mutations. We appreciate the opportunity to clarify and discuss potential reasons for this observation. Our DC registry is comprised of samples that are being sent by clinicians worldwide and therefore they do not fall under the category of routine genetic tests offered by National Health Service (NHS) in United Kingdom. As the accrual of patients increased both locally and internationally in the last decade, due to funding constraints we have been able to perform NGS on limited number of samples. For the samples where exome sequencing failed to identify candidate genes, this indicates a likely causal element in the noncoding part of the genome, which requires whole genome sequencing and further discovery research. We are actively addressing these challenges through our ongoing research activities with collaborations with broader scientific community to overcome these limitations.

In the case of patients with mutations in the POLA1 gene, the reduction in telomere length is lower than in other DC associated genes, can the authors explain or hypothesize the reason for this difference? Could this indicate that patients with POLA1 mutations may have a milder phenotype?

POLA1 primase works with the CST (CTC1-STN1-TEN1) complex in telomere replication. Similar to variable telomere lengths and diverse clinical phenotypes observed in cases with *CTC1* variants, we believe *POLA1* variants also operate in a similar fashion in causing complex DC and DCL phenotypes. We believe there is no tight correlation between disease severity and short telomeres, and therefore it is difficult to conclude that patients have mild phenotype.

1st Jul 2024

Dear Dr. Tummala,

Thank you for the submission of your revised manuscript to EMBO Molecular Medicine. Your manuscript has now been re-reviewed by two of the original reviewers. Based on their advice (included below), I am pleased to inform you that we will be able to accept your manuscript pending the following final amendments and appropriate response to reviewers:

1) Please check the "Author Checklist" carefully and complete all relevant questions. Please use the dropdowns in Column D for each question.

2) There are discrepancies in the author names between the article file and our submission system. Please fix the following discrepancies: Manthan Patel in the manuscript file vs. Manthankumar Patel in the submission system ; Reinier Raymakers in the manuscript file vs. Reiner Raymakers in the submission system ; Madapura M Pradeepa in the manuscript file vs. Pradeep Madapura in the submission system

3) In the main manuscript file, please include keywords to max. 5.

4) Please update the Data availability section describing how the data, code etc. have been made available. Please ensure that all datasets in this section are now publicly available. In addition, the Data availability section needs to be formatted according to the example below:

"The datasets and computer code produced in this study are available in the following databases:

- Chip-Seq data: Gene Expression Omnibus GSE46748 (<https://www.ncbi.nlm.nih.gov/geo/query/acc.cgi?acc=GSE46748>)

- Modeling computer scripts: GitHub (<https://github.com/SysBioChalmers/GECKO/releases/tag/v1.0>)

- [data type]: [full name of the resource] [accession number/identifier] ([doi or URL or identifiers.org/DATABASE:ACCESSION])"

5) Please rename "Conflict-of-interest disclosure" to "Disclosure and competing interests statement". We updated our journal's competing interests policy in January 2022 and request authors to consider both actual and perceived competing interests. Please review the policy <https://www.embopress.org/competing-interests> and update your competing interests if necessary.

6) Author contributions: Please remove it from the manuscript and specify author contributions in our submission system. CRediT has replaced the traditional author contributions section because it offers a systematic machine-readable author contributions format that allows for more effective research assessment. You are encouraged to use the free text boxes beneath each contributing author's name to add specific details on the author's contribution. More information is available in our guide to authors:

<https://www.embopress.org/page/journal/17574684/authorguide#authorshipguidelines>

7) References: Please correct the reference citation in the reference list. These should be alphabetically listed. Where there are more than 10 authors on a paper, note that only 10 will be listed, followed by "et al.". Please check "Author Guidelines" for more information.

<https://www.embopress.org/page/journal/17574684/authorguide#referencesformat>

8) In the Materials and Methods, please take care of the following:

- Studies with human research participants: Please update your statements on the ethics of human research participant studies to state that the experiments conformed to the principles set out in the WMA Declaration of Helsinki and the Department of Health and Human Services Belmont Report.

- Cell lines: Please be sure to include a sentence in the Materials and Methods as to whether or not the cell lines were recently authenticated and tested for mycoplasma contamination.

- Please ensure that a statement on whether or not blinding was done is included in the Materials and Methods even if no blinding was done.

- Antibodies: Please reference Appendix Table S6 in the 'Proximity ligation assay' section of the Methods so that readers may find the details of the antibodies used.

9) All Materials and Methods need to be described in the main text using our 'Structured Methods' format, which is required for all research articles. According to this format, the Methods section includes a Reagents and Tools Table (listing key reagents, experimental models, software and relevant equipment and including their sources and relevant identifiers) followed by a Methods and Protocols section describing the methods using a step-by-step protocol format. The aim is to facilitate adoption of the methodologies across labs. More information on how to adhere to this format as well as a downloadable template (.docx) for the Reagents and Tools Table can be found in our author guidelines:

<https://www.embopress.org/page/journal/17574684/authorguide#structuredmethods>

10) Please place individual sections of the manuscript in the following order: Title page - Abstract & Keywords - Introduction - Results - Discussion - Materials & Methods - Data Availability - Acknowledgements - Disclosure and Competing Interests Statement - The Paper Explained - For More Information - References - Figure Legends - Expanded View Figure Legends.

11) For the figures and figure legends, please take care of the following:

- In a routine image check, we noticed that the controls/lane markers? in EV Figure 2A are identical, yet the sizes indicated are very different. The Source Data figures provided do not include this part of the blot, so it is unclear exactly what these lanes represent. If you make changes to the figure set, please include a point-by-point describing what you have changed and why.

- Please indicate the statistical test used for data analysis in the legends of figures 5g-i; EV 2e; EV 3a.
 - Please note that the box plots need to be defined in terms of minima, maxima, centre, bounds of box and whiskers, and percentile in the legends of figures 6c; EV 3b, d-e.
 - Please note that information related to n is missing in the legends of figures 2g; 5h-i; 6c; EV 2c, e; EV 3b, d-e.
 - Please note that the error bars are not defined in the legends of figures 2g; 5h-i; EV 2c, e.
 - Please note that the scale bar is missing for figure EV 1d.
 - Please note that scale bar and its definition are missing for figure EV 2b.
- 12) Tables: Please upload all Table EV1 as one .xsl file and rename it to Dataset EV1. Please also remove its legend from the manuscript and add it to the corresponding file in a separate tab. Please also be sure to update the callouts for Table EV2 to now Table EV1 and the current Table EV1 to Dataset EV1 in main manuscript text. In addition, there is a callout for Table EV5, but there are currently only Table EV1 and Table EV2 in the submission - please correct this callout.
- 13) Appendix file: Please upload the Appendix as a single PDF (no separate image files are needed) and add page numbers separately for each of the table to the Table of Contents. The author names and affiliations may be removed.
- 14) Synopsis:
- Synopsis image: Please do not provide this figure in a Powerpoint file and rather upload it as a high-resolution jpeg file 550 pixels wide x (250-400) pixels high.
 - Synopsis text: Please remove this from the powerpoint file and upload it as a separate .doc file. Please also shorten the standfirst (maximum of 300 characters, including space).
 - Please check your synopsis text and image before submission with your revised manuscript. Please be aware that in the proof stage minor corrections only are allowed (e.g., typos).
- 15) Source Data: Please ensure that a completed Source Data checklist is uploaded, as currently the checklist is missing several check marks. Please also ensure that the Source Data are uploaded as a single source data file (zipped) per figure, with the panels clearly visible in the folder structure. Source Data for EV Fig 2B also seems to be missing.
- 16) As part of the EMBO Publications transparent editorial process initiative (see our policy here: https://www.embopress.org/transparent-process#Review_Process), EMBO Molecular Medicine will publish online a Peer Review File (PRF) to accompany accepted manuscripts. This file will be published in conjunction with your paper and will include the anonymous referee reports, your point-by-point response and all pertinent correspondence relating to the manuscript. Let us know whether you agree with the publication of the PRF and as here, if you want to remove or not any figures from it prior to publication. Please note that the Authors checklist will be published at the end of the PRF.
- 17) Please provide a point-by-point letter INCLUDING my comments as well as the reviewer's reports and your detailed responses (as Word file).

I look forward to reading a new revised version of your manuscript as soon as possible.

Yours sincerely,

Poonam Bheda

Poonam Bheda, PhD
Scientific Editor
EMBO Molecular Medicine

***** Reviewer's comments *****

Referee #1 (Remarks for Author):

The authors have considered all the point raised in the previous review satisfactorily. I have no further concerns.

Referee #2 (Comments on Novelty/Model System for Author):

I chose medium for Medical impact only because this is a relatively rare disease. One gene (POLA1) and several variants in other genes are novel, as is the functional work. This manuscript should be of great interest to the field.

Referee #2 (Remarks for Author):

The authors have responded to the majority of my prior comments/concerns. A few additional points are worthy of consideration.

1. DCL215 - the possible germline mosaic parents with two affected children. The authors could confirm paternity to rule out non-paternity.
2. POT1 - please add to the Discussion whether the heterozygous variants in these patients are similar in function/location as POT1 variants reported in patients with cancer but no features of DC.
3. Results, Methods, and Supplement - where appropriate, please clarify how many affected people were assessed in the context of the number of families.
4. Page 3 notes the use of a targeted gene panel, but I cannot find the panel in the documents. Please add as supplementary information.
5. Page 11. Team Telomere is a patient support and advocacy group, not a committee as stated.
6. Table 1. Which gnomAD version and population was used to check allele frequencies?
7. Please explain why DUT and GRHL2 are in the bar graphs on Figure 1 but not mentioned in the manuscript.

***** Reviewer's comments *****

Referee #1 (Remarks for Author):

The authors have considered all the point raised in the previous review satisfactorily. I have no further concerns.

We are thankful to this reviewer for the helpful comments and pleased to know that there are no further concerns.

Referee #2 (Comments on Novelty/Model System for Author):

I chose medium for Medical impact only because this is a relatively rare disease. One gene (POLA1) and several variants in other genes are novel, as is the functional work. This manuscript should be of great interest to the field.

Referee #2 (Remarks for Author):

The authors have responded to the majority of my prior comments/concerns. A few additional points are worthy of consideration.

1. DCL215 - the possible germline mosaic parents with two affected children. The authors could confirm paternity to rule out non-paternity.

Unfortunately, due to lack of sufficient DNA, we were not able to confirm the paternity for this family.

2. POT1 - please add to the Discussion whether the heterozygous variants in these patients are similar in function/location as POT1 variants reported in patients with cancer but no features of DC.

We have added discussion point and included 4 references that are relevant to the discussion on Page 8. The statement included reads as below:

“*POT1* variants reported here, appear to functionally mimic the ssDNA telomere binding defects (Figure 3G,H), similar to *POT1* variants described in familial cancer patients without DC or DCL features (De Boy et al, 2023; De Boy et al, 2024; Kelich et al, 2022; Robles-Espinoza et al, 2014; Shi J et al, 2014).”

3. Results, Methods, and Supplement - where appropriate, please clarify how many affected people were assessed in the context of the number of families.
We have indicated the numbers of families included in our analyses in the first section of results on Page 3. We have mentioned Patient number that are included in the analysis in the results section when describing specific figure panels.
4. Page 3 notes the use of a targeted gene panel, but I cannot find the panel in the documents. Please add as supplementary information.
The targeted gene panel information is now provided in Appendix Table S1.
5. Page 11. Team Telomere is a patient support and advocacy group, not a committee as stated.
This has now been changed to patient support and advocacy group on page 14.
6. Table 1. Which gnomAD version and population was used to check allele frequencies?
The version of gnomAD is now mentioned on Page 3.
7. Please explain why DUT and GRHL2 are in the bar graphs on Figure 1 but not mentioned in the manuscript.
We apologise for this overlook and have now mentioned both DUT and GRHL2 along with TYMS-ENOSF1 in the first section of results describing the spectrum of genetic variants on page 3 and also in Figure panel 1C.

10th Jul 2024

Dear Dr. Tummala,

Thank you for the submission of your revised manuscript to EMBO Molecular Medicine. There remain formatting issues that need to be addressed prior to acceptance of your manuscript as follows:

- 1) As previously requested, please ensure that all sequencing datasets in GEO are now publicly available. This is required prior to acceptance.
- 2) Each accession code in the Data Availability statement should have its own specific link (not a general link to the database).
- 3) As previously requested, please be sure to include a sentence in the Methods as to whether or not the cell lines were recently authenticated.
- 4) As previously requested, all materials and methods need to be described in the main text using our 'Structured Methods' format, which is required for all research articles. The description of reagents at the beginning of the Methods is not sufficient.

According to this format, the Methods section includes a Reagents and Tools Table (listing key reagents, experimental models, software and relevant equipment and including their sources and relevant identifiers) followed by a Methods and Protocols section describing the methods using a step-by-step protocol format. The aim is to facilitate adoption of the methodologies across labs. More information on how to adhere to this format as well as a downloadable template (.docx) for the Reagents and Tools Table can be found in our author guidelines:

<https://www.embopress.org/page/journal/17574684/authorguide#structuredmethods>

- 5) Please rename the "Materials & Methods" to "Methods".
- 6) The synopsis text file should include both the short standfirst (maximum of 300 characters, including space), as well as maximum 5 bullet points to summarise the key new findings. They should be designed to be complementary to the abstract - i.e. not repeat the same text. We encourage inclusion of key acronyms and quantitative information (maximum of 30 words / bullet point). Please use the passive voice.
- 7) As previously requested, please ensure that a completed Source Data checklist is uploaded
- 8) As previously requested, please also ensure that the Source Data are uploaded as a single source data file (zipped) per figure, with the panels clearly visible in the folder structure.

Please submit your revised manuscript within two weeks. I look forward to seeing a revised form of your manuscript as soon as possible.

Yours sincerely,

Poonam Bheda

Poonam Bheda, PhD
Scientific Editor
EMBO Molecular Medicine

The authors addressed the remaining editorial issues.

18th Jul 2024

Dear Dr. Tummala,

We are pleased to inform you that your manuscript is accepted for publication and is now being sent to our publisher to be included in the next available issue of EMBO Molecular Medicine. Congratulations on an excellent manuscript.

Yours sincerely,

Poonam Bheda, PhD
Scientific Editor
EMBO Molecular Medicine
